# Uncertainty Quantification in SVM prediction

## Abstract

This paper explores SVM models from the lens of uncertainty quantification (UQ), developed for regression and forecasting tasks. Unlike the Neural Network, the SVM solutions are more certain, stable, sparse, optimal and interpretable. However, there is only a limited literature addressing UQ in SVM-based prediction. At first, we provide a comprehensive summary of existing Prediction Interval (PI) estimation and probabilistic forecasting methods developed in the SVM framework. Although SVMs offer globally optimal and stable solutions, the existing literature on UQ within the SVM framework still exhibits several critical gaps. In this work, we also address these gaps and extend contemporary UQ techniques to SVMs, for promoting their applicability across diverse domains for more reliable estimation. Our major contributions include the development of sparse SVM models for PI estimation and probabilistic forecasting, an investigation of the role of feature selection in PI estimation, and the extension of SVM regression to the Conformal Regression (CR) setting to construct more stable prediction sets with finite-sample guarantees. Extensive numerical experiments highlight that SVM-based UQ methods yield PIs and probabilistic forecasts that are less uncertain than, those produced by modern complex deep learning and neural network models, particularity for small and moderate scale datasets.

## 1 Introduction

Given the training set $T' = \{(x_i, y_i) : x_i \in \mathbf{R}^n, y_i \in \mathbf{R}, i = 1, 2, ...m\}$, sampled independently from the joint distribution of the random variables $(X, Y)$, the goal of the regression task is to estimate a function that predicts the target variable $y$ based on the input variable $x$ well. However, in most applications, the prediction of the regression model may not be perfectly accurate due to the random relationship between $Y$ and $X$. For example, predicting the impact of a specific drug on a patient's heart rate based on their Body Mass Index (BMI) may not be accurate and could involve a significant degree of uncertainty. In such cases, quantifying these uncertainties is crucial for making effective decisions.

The Prediction Interval (PI) estimation is the most commonly used Uncertainty Quantification (UQ) technique in regression tasks. Given a high confidence $1 - \alpha \in (0, 1)$ and training set $T'$, the PI tube is defined as a pair of functions $(f_1, f_2)$. It is said to be well calibrated if it satisfies $P(f_1(X) \leq Y \leq f_2(X)|X) \geq 1 - \alpha$. The objective of the PI models is to obtain a PI tube with the minimum possible width while ensuring the target calibration. Therefore, the performance of a PI estimation method is mainly evaluated using two criteria: Prediction Interval Coverage Probability (PICP), which computes the fraction of $y$ values within the PI tube, and Mean Prediction Interval Width (MPIW), quantifying the width of the PI tube.

The PI estimation models explore the different characteristics of the conditional distribution $Y|X$, rather than focusing only on $E(Y|X)$, as done in standard regression tasks. The basic approach of PI models involves estimating a pair of quantile functions (Koenker & Bassett Jr (1978)), say $(f_q(x), f_{1+q-\alpha}(x))$, of the conditional distribution $Y|X$, for some $0 \leq q \leq \alpha$, where the $q^{th}$ quantile function for given $x$ is $f_q(x)$, that should be infimum over all functions satisfying $P(y \leq f(x)|x) = q$.

For time-series data, estimating the PI for future observations using an auto-regressive approach is referred to as probabilistic forecasting. Both PI estimation and probabilistic forecasting models are widely investigated in the Neural Network (NN) architectures in the literature. The PI estimation methods in the NN literature

can primarily be divided into two main categories. A popular class of PI estimation methods assumes that the conditional distribution $Y|X$ follows a particular distribution (often normal) and obtains the quantile functions by computing the inverse of the corresponding Cumulative Distribution Function(CDF). Some important of them are Bayesian method (MacKay (1992); Bishop (1995)), Delta method (De VlEAUX et al. (1998); Hwang & Ding (1997); Seber & Seber (2015)) and Mean Variation Estimation (MVE) method (Nix & Weigend (1994)). Some of the recent NN architectures for the probabilistic forecasting task with distribution assumptions are Mix Density Network (Bishop (1994); Zhang et al. (2020a)), Deep Auto-regressive Network (Deep AR) (Salinas et al. (2020)).

The other class of PI estimation and probabilistic forecasting methods believe in estimating the pair of quantile functions $(f_q(x), f_{1+q-\alpha}(x))$ in a distribution-free setting without imposing any assumptions regarding the distribution of $Y|X$. For estimation of the $q^{th}$ quantile function, $f_q(x)$, most of them minimize the pinball loss function (Koenker & Bassett Jr (1978)). The pinball loss-based NN model, also known as Quantile Regression Neural Network (QRNN) (Taylor (2000); Cannon (2011)) is the main PI estimation method, which has been utilized in various engineering applications. The pinball loss-based NN model has been frequently applied to probabilistic forecasting of wind (Wan et al. (2016)), electric load (Zhang et al. (2018; 2020b)), electric consumption (He et al. (2019)), flood risks (Pasche & Engelke (2024)) and solar energy (Lauret et al. (2017)). Some of the distribution-free PI estimation NN methods consider the minimization of a particularly designed loss function for the direct and simultaneous estimation of the bounds of the PI. The most prominent among these are the Lower–Upper Bound Estimation (LUBE) neural network, the Quality-Driven (QD) loss neural network Pearce et al. (2018), and the Tube-loss neural network Anand et al. (2024).

However, a well-calibrated High Quality (HQ) PI guarantees the target coverage level $t$ only asymptotically, and may fail to achieve it on finite test samples. In real-world decision-making, especially in high-stakes applications, finite test samples coverage guarantees are often essential. Conformal Regression (CR) (Vovk et al. (1999; 2005)) provides a principled framework through which PI models can be adapted to ensure such finite-sample coverage guarantees, making them more suitable for practical deployment.

The aforementioned UQ techniques derive a pair of quantile functions to quantify the uncertainty in the relationship between $X$ and $Y$, However, their estimates themselves may also involve a certain degree of uncertainty. Therefore, researchers typically consider two main sources of uncertainty when assessing the overall uncertainty: data uncertainty (aleatoric uncertainty) and model uncertainty (epistemic uncertainty). Data uncertainty arises from the inherent noise or variability in the relationship between $X$ and $Y$ while model uncertainty primarily arises from the uncertainty in the estimated model parameters.

For the UQ task in NN and deep learning regression, accurately capturing model uncertainty, in addition to data uncertainty is essential. This is due to the significant variability in the learned weights, as the underlying objective functions are typically non-convex. As a result, training the same model multiple times with the same dataset and hyperparameter settings can still lead to different local minima. Studies by Lakshminarayanan et al. (2017) and Pearce et al. (2018) have shown that ensemble methods are among the most effective approaches for addressing model uncertainty in NN. The key idea is to train the model multiple times with different random initializations on the same training set $T'$, and then estimate model uncertainty by measuring the variability in the resulting learned parameters.

However, in contrast to NN, SVM does not suffer from such parameter uncertainty, as they obtain the global optimal solution that remains invariant to the choice of the initialization. It makes the SVM more trustworthy and less uncertain than NN. Apart from the global optima, the SVM also offers simple, interpretable and often sparse solutions along with an explicit mechanism to incorporate the regularization.

Despite these notable advantages, SVM remains surprisingly almost under-explored from the lens of the UQ. In contrast to detailed NN literature, only a limited number of SVM-based approaches have been proposed for PI estimation and probabilistic forecasting tasks. This paper provides a comprehensive survey of SVM models developed for UQ tasks. Despite the advantages of global optimal and stable solution, the existing SVM literature on UQ contain critical gaps. In this paper, we also bridge these gaps and further extend contemporary UQ techniques to the SVM framework to promote the SVM UQ application across diverse domains for a more reliable estimate.

We summarize the key contributions of this work as follows

(a) First, we carefully review the existing literature on PI estimation and probabilistic forecasting methods in SVM. We outline the desirable properties of an ideal PI model and compare the PI estimation and probabilistic forecasting methods in SVM against them. We find that only two of SVM PI methods attain the global optimal solution but, none of them achieve a sparse solution vector.

(b) Building on this motivation, we develop a sparse SVM solution for PI estimation and probabilistic forecasting tasks by simply solving a pair of pinball loss problems with $L_1$ regularization via Linear Programming Problem (LPP). Our Sparse SVM PI model preserves the classical properties of SVM by achieving both a globally optimal and sparse solution. Unlike NN, the global optimal solution of Sparse SVM PI formulation ensures the zero parameter uncertainty while its sparse solution reduces the complexity of the solution. It enables the sparse SVM PI method to estimate a less uncertain and complex PI.

(c) Further, we investigate the importance of feature selection in PI estimation particularly in high dimensional regression tasks. At first, we perform the feature selection task by developing a simple yet effective feature selection algorithm for PI estimation using our sparse SVM PI model. We show that our algorithm not only successfully discards a significant percentage of features but, also improves the quality of the PI while learning the PI for high-dimensional data. To the best of our knowledge, there is no existing work that studies the feature selection problem in the context of PI estimation.

(d) Finally, we note that although SVM models inherently exhibit zero model uncertainty, they have not yet been extended to the conformal framework for regression tasks. In this work, we bridge this gap by adapting SVM regression to the conformal prediction setting, thereby ensuring finite-sample coverage guarantees. Compared to NN, we show that the conformal prediction sets produced by SVM model are more stable, certain and interpretable due to its global optimal solution.

(e) We conduct extensive experiments on both artificial and real-world benchmark datasets to empirically verify the above claims. We show that our Sparse SVM PI model not only yields sparse solutions but also tends to produce slightly improved PI quality compared to existing SVM-based PI models (cf. Tables 2–7). We show the advantages of the feature selection method in the PI estimation task on benchmark high-dimensional datasets using our Sparse SVM-based PI model (cf. Table 10-11). We empirically verify that the SVM based CR model solution is more stable, certain than NN based CR model (cf. Table 14). Finally, we conduct experiments demonstrating that SVM-based probabilistic forecasting models can produce comparable and in some cases superior estimates of data uncertainty, measured using PICP and MPIW, when compared with recent deep probabilistic forecasting models that require training with significantly larger numbers of parameters (cf. Tables 12–13). The results also reveal that modern deep probabilistic forecasting models exhibit substantial model uncertainty, whereas the proposed SVM-based models achieve stable predictions with no observable model uncertainty.

The remainder of this paper is structured as follows. Section 2 provides a systematic review of the prerequisite concepts required for understanding of the SVM models and UQ techniques. Section 3 presents a detailed discussion of existing SVM models for PI estimation and probabilistic forecasting, and concludes that only two SVM-based PI methods, namely SVQR and LS-SVR based PI models, attain globally optimal solutions. This section also highlights the importance and necessity of sparse solutions in PI estimation. In Section 4, we introduce Sparse SVM algorithms for PI estimation by solving a pair of LPPs, which yield globally optimal and sparse solutions with zero model uncertainty. Algorithm 3 in Section 4 presents the Sparse SVM–based feature selection method, while Algorithm 4 extends the SVM-based PI model to the conformal regression (CR) framework. Section 5 reports the numerical results supporting our claims, and Section 6 concludes the paper by summarizing the contributions and outlining future research directions.

## 2 Preliminaries

In this section, we outline key concepts and methods relevant to PI estimation techniques in SVM.

### 2.1 Quantile Regression and SVM

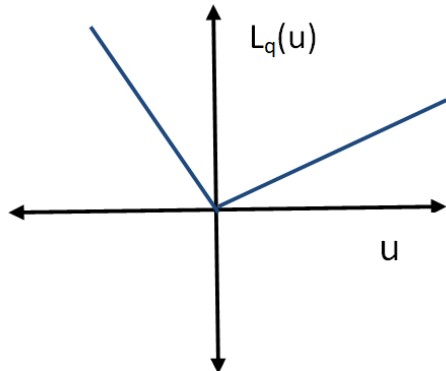

Figure 1: Pinball loss function

In distribution-free setting, for a given quantile $q \in (0, 1)$, the quantile value is estimated by minimizing the pinball loss function given by

$$\rho_q(u) = \begin{cases} qu, & \text{if } u \geq 0, \\ (q-1)u, & \text{Otherwise.}, \end{cases} \tag{1}$$

For the estimation of the conditional quantile function, $u$ represents the error obtained by subtracting the estimates $f(x_i)$ from its target values $y_i$. For a given quantile $q \in (0, 1)$, training set $T' = \{(x_i, y_i) : x_i \in \mathbf{R}^n, y_i \in \mathbf{R}, i = 1, 2, ...m\}$ and class of function $F$, let us suppose that $f_T$ is the solution of the problem $\min_{f \in F} \sum_{i=1}^{m} \rho_q(y - f(x_i))$. Takeuchi et al., have shown that the fraction of $y$ values lying below the function $f_T(x)$ is bounded from above by $qm$ and asymptotically equals $qm$ with probability 1 under a very general condition in their work (Takeuchi et al. (2006)).

Given the training set, SVM models estimate the function in the form of $f(x) = w^T \phi(x) + b$, where $\phi$ maps the input variable $x$ into the high dimensional feature space, such that for any pair of $x_i$ and $x_j$ in $\mathbf{R}^n$, $\phi(x_i)^T \phi(x_j)$ can be obtained by the well defined kernel function $k(x_i, x_j)$. By the use of the kernel trick and representer theorem Schölkopf et al. (2001), the SVM estimate $f(x) = w^T \phi(x) + b$ can be represented by the kernel generated function in the form of $\sum_{i=1}^{m} k(x_i, x)u_i + b$, where $k$ is positive-semi definite kernel Mercer (1909). This representation eliminates the need for explicit knowledge of the mapping $\phi$.

### 2.2 Support Vector Quantile Regression model

The Support Vector Quantile Regression (SVQR) model (Takeuchi et al. (2006)) minimizes the $L_2$-norm of the regularization along with the empirical risk computed by the pinball loss function. For $q^{th}$ quantile function estimation, it seeks the solution of the problem

$$\min_{(w,b)} \frac{1}{2} w^T w + C \sum_{i=1}^{m} \rho_q(y_i - (w^T \phi(x_i) + b)), \tag{2}$$

which can be equivalently converted to the following Quadratic Programming Problem (QPP)

$$\min_{(w,b,\xi,\xi^*)} \frac{1}{2} w^T w + C \sum_{i=1}^{m} (q\xi_i + (1-q)\xi_i^*)$$

subject to,

$$y_i - (w^T \phi(x_i) + b) \leq \xi_i,$$
$$(w^T \phi(x_i) + b) - y_i \leq \xi_i^*,$$
$$\xi_i, \xi_i^* \geq 0, \ i = 1, 2, ..m, \quad (3)$$

where $C \geq 0$ is the user defined parameter for trading off the empirical risk against the model complexity.

To efficiently solve QPP (3), we often focus on obtaining the solution to its corresponding Wolfe dual problem, which is given by

$$\min_{(\alpha,\beta)} \sum_{i=1}^{m} \sum_{j=1}^{m} (\alpha_i - \beta_i) k(x_i, x_j)(\alpha_j - \beta_j) - \sum_{i=1}^{m} (\alpha_i - \beta_i) y_i$$

subject to,

$$\sum_{i=1}^{m} (\alpha_i - \beta_i) = 0,$$
$$0 \leq \alpha_i \leq Cq, \ i = 1, 2, ..m,$$
$$0 \leq \beta_i \leq C(1-q), \ i = 1, 2, ..m, \quad (4)$$

where $(\alpha_i, \beta_i), i = 1, 2, .., m$, are Lagrangian multipliers.

After obtaining the optimal solution of the dual problem (4), $(\alpha_i^*, \beta_i^*), i = 1, 2, .., m$, the $q^{th}$ quantile function is estimated by

$$f_q(x) = \sum_{i=1}^{m} (\alpha_i^* - \beta_i^*) k(x_i, x) + b. \quad (5)$$

The estimation of the bias term $b$ can be obtained by using the KKT conditions of the primal problem (3). For this, we need to select the every training point $(x_k, y_k)$ which corresponds to $0 < \alpha_k^* < Cq$ or $0 < \beta_k^* < C(1-q)$ and compute

$$b_k = y_k - \sum_{i=1}^{m} (\alpha_i^* - \beta_i^*) k(x_i, x_k). \quad (6)$$

The final value of bias $b$ can be obtained by computing the mean of all $b_k$.

## 2.3 Least Squares Support Vector Regression

For estimating the mean regression using training set $T'$, the LS-SVR model Suykens et al. (2002) minimizes the least square loss function along with the $L_2$-norm of regularization in the following problem.

$$\min_{(w,b,\xi)} \frac{1}{2} w^T w + C \sum_{i=1}^{m} (\xi_i^2)$$

subject to,

$$y_i - (w^T \phi(x) + b) = \xi_i, \ i = 1, 2, ..m. \quad (7)$$

The solution of problem (7) can be obtained through the following system of equations

$$\begin{bmatrix} 0 & e^T \\ e & K(A, A^T) + \frac{2}{C}I \end{bmatrix} \begin{bmatrix} b \\ \alpha \end{bmatrix} = \begin{bmatrix} 0 \\ Y \end{bmatrix}, \quad (8)$$

where $K(A, A^T)$ is an $m \times m$ kernel matrix constructed from the training set $T'$, $e$ is an $m$-dimensional column vector of ones, and $I$ represents the $m \times m$ identity matrix. After obtaining the $(b, \alpha)$ from (8), the

LS-SVR estimate the regression function for a given $x \in \mathbb{R}^n$ using

$$f(x) = \sum_{i=1}^{m} k(x_i, x)\alpha_i + b. \tag{9}$$

## 2.4 Probabilistic Forecasting

The task of probabilistic forecasting is essentially an extension an extension of the PI estimation in an auto-regressive setting. Consider the time series observations $T = \{x_1, x_2, ...., x_t\}$, recorded at $t$ different time stamps. If $p < t$ denotes the effective lag window, then auto-regressive models estimate the relationship between $z_i := (x_{i-p+1}, \ldots, x_i)$ and $x_{i+1}$ for $i = p, p+1, \ldots, t-1$ using the training set $T'$, and use this learned relationship to forecast future observations. Point forecasting models aim to estimate the conditional expectation $E(x_{i+1} \mid z_i)$. However, in many cases, such forecasts may incur significant errors due to inherent noise and volatility in the data. Probabilistic forecasting quantifies this uncertainty by estimating PI.

The task of probabilistic forecasting is to estimate the PI for $x_{i+1}$ given the input $z_i$ for $i \geq t$. The SVM based probabilistic forecasting models obtain the estimate of the PI $[\hat{F}_q(z_i), \hat{F}_{1+q-\alpha}(z_i)]$, where $\hat{F}_q(z_i)$ and $\hat{F}_{1+q-\alpha}(z_i)$ are kernel-generated functions, estimating the $q^{th}$ and $(1 - \alpha + q)^{th}$ quantiles of the conditional distribution $(x_{i+1} \mid z_i)$ for some $q \in (0, \alpha)$. Distribution-free probabilistic forecasting methods estimate quantile functions directly, without making any assumptions about the conditional distribution $(x_{i+1} \mid z_i)$.

## 2.5 Conformal Regression

Conformal Regression (CR) (Vovk et al. (1999; 2005)) provides a general framework for adjusting PI models to guarantee the target coverage $1 - \alpha$ on finite test samples, assuming only that the data are exchangeable.

The split CR approach (Papadopoulos et al. (2002); Papadopoulos (2008)) starts by dividing the available training data $T$ into two separate subsets: a training set $I_1$ used to fit the predictive model, and a calibration set $I_2$ used for PI adjustment. A nonconformity score function is then introduced to quantify the disagreement between predicted value for $y_i$, given input $x_i$ and its actual observed value. These nonconformity scores are computed on the calibration set $I_2$. A obvious choice of the non-conformity score is the absolute residual, computed by $|y_i - \hat{f}(x_i)|, \quad i \in I_2$, where $\hat{f}$ is the estimate of the mean regression model trained on $I_1$.

Romano et al. have developed the quantile regression based nonconformity score for obtaining the fully adaptive conformal prediction set in their work Conformalized Quantile Regression (CQR) (Romano et al. (2019)), which is given by

$$E_i = \max\{\hat{F}_{lo}(x) - y_i, \ y_i - \hat{F}_{hi}(x)\}, \tag{10}$$

where $\hat{F}_{lo}(x)$ and $\hat{F}_{hi}(x)$ are the estimates of the $q^{th}$ and $(1 + q - \alpha)^{th}$ quantile function on set $I_1$ for some $0 \leq q \leq \alpha$.

After computing the non-conformity scores, the CR methodology requires the computation of $(1-\alpha)(1+\frac{1}{|I_2|})$-th empirical quantile of the non-conformity score. In case of CQR, we denote it with $Q_{1-\alpha}(E_i, I_2)$ and obtain the prediction set for a new test point $x_{m+1}$ as

$$C(x_{m+1}) = [\hat{F}_{lo}(x_{m+1}) - Q_{(1-\alpha)}(E, I_2), \hat{F}_{hi}(x_{m+1}) + Q_{1-\alpha}(E, I_2)] \tag{11}$$

# 3 PI estimation in SVM

In this section, we gather in detail the PI estimation and probabilistic forecasting methods developed in the SVM literature and compare their advantages and limitations.

## 3.1 PI estimation through LS-SVR

One of the naive PI estimation methods in SVM literature follows the normal assumption regarding the distribution of $Y|X$ and estimates its mean through (9) by training the LS-SVR model. The error distribution

$\epsilon_i = y_i - f(x_i)$ follows a normal distribution with zero mean and variance $\sigma$. This variance can be estimated from the error computed on the training set $T'$. The pair of quantile bounds required for PI is estimated as $(f(\hat{x}) + \epsilon_{\frac{\alpha}{2}}, f(\hat{x}) + \epsilon_{1-\frac{\alpha}{2}})$, where $\epsilon_q$ is the $q^{th}$ quantile of the error. A more refined and bias-corrected PI based on the LS-SVR model was proposed in (De Brabanter et al. (2010); Cheng et al. (2014)).

### 3.2 PI estimation through SVQR

Given the target confidence $1 - \alpha$ with training set $T'$, the PI model requires the estimation of the pair of quantile functions $(f_q(x), f_{1+q-\alpha}(x))$ of the conditional distribution $Y/X$. The SVQR model can be trained twice for the estimation of this pair of quantile functions for some $0 \leq q \leq \alpha$. We detail the algorithm for PI estimation through SVQR in Algorithm 1. In Algorithm 1, the tuning of $C$ refers to selecting the value of $C$ from a specified range such that the SVQR estimate obtains the least coverage error.

---

**Algorithm 1** PI estimation through SVQR

1: **procedure** PI THROUGH SVQR$(T, 1 - \alpha)$
2:     Choose some $\bar{q} \in (0, 1 - \alpha)$
3:     **for** each $q \in \{\bar{q}, (1 + \bar{q} - \alpha)\}$ **do**
4:         Solve the QPP problem (4) by tuning the value of $C$. Obtain the solution $(\alpha^*, \beta^*)$.
5:         Estimate the function $f_q(x)$ using (5).
6:     **return** $(f_{\bar{q}}(x), f_{1+\bar{q}-\alpha}(x))$

---

A key challenge in estimating prediction intervals (PI) using the quantile approach is to determine a good choice of $\bar{q}$ for obtaining the narrower PI. For a symmetric noise distribution, $\bar{q} = \frac{\alpha}{2}$ is expected to produce the PI with minimum width. However, this does not hold for an asymmetric noise distribution. In the latter case, $\bar{q}$ should be selected such that the resulting PI passes through the denser regions of the data cloud. Furthermore, for each choice of $\bar{q}$, the optimization problem (4) must be solved twice for $q = \bar{q}$ and $q = 1 + \bar{q} - \alpha$ to obtain the prediction interval (PI). This increas es the overall computational complexity of the PI estimation process, making it both time-consuming and challenging in practice.

To simplify the PI estimation process, researchers have developed direct PI estimation methods, which solve a single optimization problem to obtain the both bounds of PI simultaneously. These methods are designed with a specialized loss function that can be minimized to obtain the both bounds of PI simultaneously. Some important of them are LUBE loss (Khosravi et al. (2010)), Quality-Driven (QD) loss (Pearce et al. (2018)) and Tube loss (Anand et al. (2024)) functions. We describe those also formulated within the SVM framework as follows.

### 3.3 PI estimation through Tube loss

(Anand et al. (2024)) have developed the Tube loss for PI estimation and probabilistic forecasting. It can be minimized directly to obtain the bounds of the PI simultaneously. The minimizer of the Tube loss function guarantees the target coverage $1 - \alpha$ asymptotically. Also, the PI tube can also be shifted up or down by tuning its parameter $r$ so that it passes through the denser region of data cloud for minimal PI width. Furthermore, the width of the PI tube can be explicitly minimized in its optimization problem through the parameter $\delta$. It helps to improve the PI width, when the PI tube achieves a coverage higher than the target on the validation set.

The Tube loss function is a kind of two-dimensional extension of the pinball loss function (Koenker & Bassett Jr (1978)). For a given $1 - \alpha \in (0, 1)$ and $u_2 \leq u_1$, the Tube loss function is given by

$$\rho_{1-\alpha}^r(u_2, u_1) = \begin{cases} (1-\alpha)u_2, & \text{if } u_2 > 0, \\ -\alpha u_2, & \text{if } u_2 \leq 0, u_1 \geq 0 \text{ and } ru_2 + (1-r)u_1 \geq 0, \\ \alpha u_1, & \text{if } u_2 \leq 0, u_1 \geq 0 \text{ and } ru_2 + (1-r)u_1 < 0, \\ -(1-\alpha)u_1, & \text{if } u_1 < 0, \end{cases} \tag{12}$$

where $0 < r < 1$ is a user-defined parameter and $(u_2, u_1)$ are errors, representing the deviations of $y$ values from the bounds of PI.

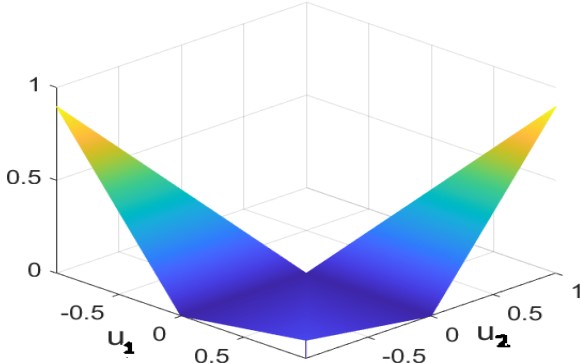

Figure 2: Tube loss function for $1 - \alpha = 0.9$.

Figure 2 illustrates the Tube loss for $(1 - \alpha) = 0.9$ with $r = 0.5$. For $r = 0.5$, the Tube loss function is a continuous loss function of $u_1$ and $u_2$, symmetrically positioned around the line $u_1 + u_2 = 0$. In all experiments with a symmetric noise distribution, the $r$ parameter in the Tube loss function should be set to 0.5 to capture the denser region of $y$ values.

The Tube loss SVM model seeks a pair of kernel-generated functions

$$\mu_1(x) = \sum_{i=1}^{m} k(x_i, x)\alpha_i + b_1 \text{ and } \mu_2(x) = \sum_{i=1}^{m} k(x_i, x)\beta_i + b_2 \tag{13}$$

by minimizing the optimization problem

$$\min_{(\alpha, \beta, b_1, b_2)} J(\alpha, \beta, b_1, b_2) = \frac{\lambda}{2}(\alpha^T \alpha + \beta^T \beta) + \sum_{i=1}^{m} \rho_{1-\alpha}^{r}\big(y_i, \big(K(A^T, x_i)\alpha + b_1\big), \big(K(A^T, x_i)\beta + b_2\big)\big)$$

$$+ \delta \sum_{i=1}^{m} \big|(K(A^T, x_i)(\alpha - \beta) + (b_1 - b_2)\big|, \tag{14}$$

where $\delta, r$ and $\lambda$ are user-defined parameters and $A$ is the $m \times n$ data matrix containing the training set. Further, details on the Tube SVM problem and its minimization using the gradient descent method can be found in (Anand et al. (2024)).

### 3.4   PI estimation through LUBE loss

The LUBE method (Khosravi et al. (2010)) was originally developed in the NN framework but, was extended to the SVM framework later in (Shrivastava et al. (2014; 2015)) for probabilistic forecasting of electric prices. For the given target confidence $(1 - \alpha)$, and training set $T'$, the LUBE SVM model seeks a pair of kernel generated functions of (13), $\mu_1(x)$ and $\mu_2(x)$, which are obtained by minimizing the following loss function

$$CWC = \frac{1}{R} MPIW \big(1 + \gamma \, (PICP)e^{-\eta(PICP-(1-\alpha))}\big). \tag{15}$$

Here, MPIW is the average width of estimated PI on training set, computed by $\frac{1}{m}\sum_{i=1}^{m}(\mu_2(x_i) - \mu_1(x_i))$. As discussed earlier, PICP is the coverage of the estimated PI and computed using the $PICP = \frac{1}{m}\sum_{i=1}^{m} k_i$, where

$$k_i = \begin{cases} 1, & \text{if } y_i \in [\mu_1(x), \mu_2(x)]. \\ 0, & \text{Otherwise.} \end{cases}$$

|  | LS-SVR PI | SVQR PI | LUBE | Tube loss |
|---|---|---|---|---|
| Distribution-free method | No | Yes | Yes | Yes |
| Asymptotic guarantees | Only with normal noise | Yes | No | Yes |
| Direct PI estimation | Yes | No | Yes | Yes |
| PI tube movement | No | Yes | No | Yes |
| Global optimal solution | Yes | Yes | No | No |
| Re-calibration | No | No | Yes | Yes |
| Sparsity | No | No | No | No |

Table 1: Comparisons of the Quantile, LUBE, QD and Tube loss based PI estimation models

Further, the $\gamma(PICP) = \begin{cases} 0, & \text{if } PICP \geq 1 - \alpha, \\ 1, & \text{otherwise}, \end{cases}$ , $R$ is the range of response values $y_i$ and $\eta$ is the user-defined parameter.

The major problem with the LUBE cost function (15) is that it is very difficult to be optimized because, the PICP is a step function. Khosravi et al. have solved the LUBE cost function (15) using the Particle Swarm Optimization (PSO) (Kennedy & Eberhart (1995)) to estimate the PI. However, due to sub-optimal solution, high-quality PI is not always observed. Pearce et al. refined the LUBE cost function and use a sigmoidal function to approximate the PICP, allowing the application of the gradient descent method for training the NN for PI estimating in their work (Pearce et al. (2018)).

## 3.5 Comparison of PI estimation SVM models

In Table 1, we visualize the desirable properties for a PI estimation model and compare the SVM methods in light of them with a detailed discussion as follows.

(a) Distribution-free method : - The LS-SVR PI model assumes that the underlying noise distribution of the data is normal and may obtain poor estimates otherwise. In the literature, distribution-based PI methods often struggle to achieve consistent performance across various datasets. The SVQR, LUBE, and Tube loss methods provide PI estimation without assuming any specific distribution, allowing them to generate high-quality PI even in the presence of non-normal noise.

(b) Asymptotic guarantees:- A fundamental requirement in PI estimation models is that the obtained PI should guarantee the target coverage $1 - \alpha$ at least asymptotically, which remain missing in the LUBE method. The LS-SVR PI model provides this guarantee only in presence of normal noise. The SVQR and Tube loss based PI methods provide this asymptotic guarantee. The asymptotic guarantee of the SVQR based PI model is implied from the asymptotic guarantee of the pinball loss function minimizer for quantile regression detailed in (Takeuchi et al. (2006)).

(c) Direct PI estimation:- As detailed in Algorithm 1, the SVQR PI model obtains the two bounds of PI by solving pair of SVQR problems one by one. The LUBE and Tube loss-based PI models simultaneously obtain the bounds of the PI by solving a single optimization problem.

(d) PI tube movement:- The PI tube movement is one another important desirable feature for PI estimation. This movement allows the PI to pass through the denser regions of the data cloud, helping to minimize the width of the PI, while achieving the target coverage. The centered PI is ideal only in the presence of the symmetric noise. However, in the presence of asymmetric noise in the data, the width of the prediction interval (PI) can be reduced by shifting it upward or downward without compromising its coverage. The SVQR and Tube loss-based PI models enable PI movement through their parameter $\hat{q}$ and $r$ respectively.

(e) Global Optimal Solution:- One of the attractive features of the standard SVM methods is that they guarantee a global optimal solution by solving a convex optimization problem. However, in PI estimation, only the SVQR and LS-SVR based PI models maintain this guarantee by minimizing

the convex loss function in its optimization problems. The Tube loss problem (14) is non-convex and hence fails to guarantee the global optimal solution. Furthermore, the LUBE problem (15) is highly discontinuous and relies on meta-heuristic algorithms for its solution. It often makes the LUBE solution suboptimal, resulting in poor PI quality.

(f) Re-calibration:- As detailed in Algorithm 1, the SVQR PI model obtains the bound of PI by solving the pair of SVQR problems one-by one. It can not explicitly minimize the width of PI in its optimization problem. This limitation prevents SVQR PI from using the recalibration feature. In recalibration, PI models are retrained to reduce interval width, when empirical coverage obtained on validation set exceeds the target $1 - \alpha$ significantly. During retraining, the PI models increase the value of the parameter ($\delta$ in case of the Tube loss) that trade-off the width of the PI against the coverage in the optimization problem. The LUBE and Tube loss-based models explicitly incorporate the minimization of prediction interval (PI) width in their problems, thereby enabling recalibration, which is practically useful for further reducing the PI width.

(g) Sparsity:- Sparsity is yet another promising feature offered by the initial SVM models developed for the classification and regression. However, it remains missing with all PI estimation models developed in the SVM framework.

Only few of components of a sparse vector are non-zero. A sparse solution in a learning model is desirable and important as it offers significant advantages. In case of the linear model, it can help us to drop irrelevant features making the learning task more simple and effective. Further, in case of learning non-linear function also, it reduces the complexity of the model without compromising its predictive ability on train set that makes the model own better generalization ability on unseen test points.

In the SVM literature, two primary approaches are commonly used to obtain sparse solutions, which often lead to confusion. The first approach minimizes the empirical risk with an additional $L_1$-norm regularization term, while the second relies on the $\epsilon$-insensitive loss function to minimize empirical error. However, when estimating a linear regression model of the form $w^T x + b$, with $w \in \mathbb{R}^n$, the latter approach can only induce sparsity in the dual variables (Lagrangian multipliers) and does not guarantee sparsity in $w$. As a result, it cannot eliminate irrelevant features during the learning process. In contrast, explicitly minimizing the $L_1$-norm of $w \in \mathbb{R}^n$ within the optimization problem directly enforces sparsity in $w$, thereby enabling feature selection.

## 4 Sparse PI estimation in SVM

In this section, we develop the Sparse Support Vector Quantile Regression (SSVQR) based PI estimation model. The SSVQR PI model inherits all properties of SVQR listed in Table 1 and also brings the sparse solution as well. Additionally, it addresses the feature selection problem in PI estimation efficiently and also can obtain the sparse estimates in CR setting.

In literature, there are few works that obtain the sparse solution while minimizing the pinball loss function or its variants. Some of the literature such as (Taylor (2000); Rastogi et al. (2018)) obtains the sparse solution for the classification task in the SVM framework while minimizing the variant of the pinball loss function. In (Tanveer et al. (2021)), authors have obtained the sparse solution by minimizing a variant of the pinball loss for clustering problem. For quantile estimation, there are a few literature like (Anand et al. (2020); Ye et al. (2025)) which attempt to develop the $\epsilon$-insensitive variant of the pinball loss function, inspired by the $\epsilon$-insensitive loss function used in standard SVM regression (Vapnik (2013)).

However, as discussed in above section, the minimization of the $\epsilon$-insensitive loss function can not directly eliminate the features. In light of the above, we minimize a pair of pinball loss functions with an added $L_1$-norm regularization to obtain a sparse SVM solution for the PI estimation task. Li & Zhu (2008) have also employed the pinball loss function with $L_1$-norm regularization for sparse quantile estimation. However, for a stable solution, we reformulate the problem as a linear programming problem (LPP) and further extend it to address feature selection in PI estimation.

### 4.1 Sparse Support Vector Quantile Regression model

The SSVQR minimizes the pinball loss function with $L_1$- norm regularization. It seeks the solution of the following problem

$$\min_{(w,b)} \frac{1}{2}||w||_1 + C\sum_{i=1}^{m} \rho_q(y_i - (w^T\phi(x_i) + b)), \tag{16}$$

where $C \geq 0$ is the user defined parameter for trading of the regularization against the empirical loss. The solution to problem (16) is sparse, similar to LASSO regression, as the minimization of the $L_1$-norm regularization compels the weight coefficients to shrink to zero.

With the help of the kernel trick, the representor theorem rewrites the estimated function $f(x) = (w^T\phi(x)+b)$ as $\sum_{j=1}^{m} k(x_j,x)u_j + b$. Also, the pinball loss $\rho_q(u)$ in (1) is equivalent to $q\max(u,0) + (1-q)\max(-u,0)$.

Further, let us consider $u_i = y_i - \left(\sum_{j=1}^{m} k(x_j,x_i)u_j + b\right)$ and two slack variables $\xi_i = \max(u_i,0)$ and $\xi_i^* = \max(-u_i,0)$ to measure the error corresponding to the data points lying above and lower side of the regression function respectively.

Now, the problem (16) can be written as

$$\min_{(u,b)} \frac{1}{2}||u||_1 + C\sum_{i=1}^{m}(q\xi_i + (1-q)\xi_i^*),$$

subject to,

$$\xi_i = \max\left(y_i - \left(\sum_{j=1}^{m} k(x_j,x_i)u_j + b\right),0\right), i = 1,2,...m,$$

$$\xi_i^* = \max\left(\left(\sum_{j=1}^{m} k(x_j,x_i)u_j + b\right) - y_i,0\right), i = 1,2,...m,$$

that can be simplified as

$$\min_{(u,b,\xi,\xi^*)} \frac{1}{2}||u||_1 + C\sum_{i=1}^{m}(q\xi_i + (1-q)\xi_i^*)$$

subject to,

$$\xi_i \geq y_i - \left(\sum_{j=1}^{m} k(x_j,x_i)u_j + b\right), \quad \xi_i \geq 0, \ i = 1,2,..m,$$

$$\xi_i^* \geq \left(\sum_{j=1}^{m} k(x_j,x_i)u_j + b\right) - y_i, \quad \xi_i^* \geq 0, \ i = 1,2,..m,$$

which can be written as

$$\min_{(u,b,\xi,\xi^*)} \frac{1}{2}||u||_1 + C\sum_{i=1}^{m}(q\xi_i + (1-q)\xi_i^*)$$

subject to,

$$y_i - \left(\sum_{j=1}^{m} k(x_j,x_i)u_j + b\right) \leq \xi_i,$$

$$\left(\sum_{j=1}^{m} k(x_j,x_i)u_j + b\right) - y_i \leq \xi_i^*,$$

$$\xi_i,\xi_i^* \geq 0, \ i = 1,2,..m. \tag{17}$$

Without loss of generality, let us consider the solution vector $u = r - p$, where $r$ and $p$ are vectors of positive numbers i,e., $r_i, p_i > 0, i = 1,2,..,m$ , then similar to the (Mangasarian et al. (2006)), the problem (17) can

be expressed as

$$\min_{(r,p,b,\xi,\xi^*)} \frac{1}{2} \sum_{i=1}^{m} (r_i + p_i) + C \sum_{i=1}^{m} (q\xi_i + (1-q)\xi_i^*)$$

subject to,

$$y_i - \left( \sum_{j=1}^{m} k(x_j, x_i)(r_j - p_j) + b \right) \le \xi_i,$$

$$\left( \sum_{j=1}^{m} k(x_j, x_i)(r_j - p_j) + b \right) - y_i \le \xi_i^*,$$

$$\xi_i, \xi_i^*, r_i, p_i \ge 0, \ i = 1, 2, ..m. \tag{18}$$

The above problem (18) is a LPP with $4m$ variables, $2m$ linear constraints and $4m$ non-negative constraints, which can be efficiently solved by any LPP solver. The optimal solution $(r^*, p^*, b^*)$ of the LPP (18) determines the estimate of the $q^{th}$ quantile function using

$$f_q(x) = \sum_{i=1}^{m} (r_i^* - p_i^*) k(x_i, x) + b. \tag{19}$$

The asymptotical properties of the SSVQR model remain similar to the SVQR model detailed in (Takeuchi et al. (2006)).

## 4.2   PI estimation through SSVQR

We detail the algorithm for PI estimation through SSVQR in 2.The SSVQR PI preserves the properties of the SVQR PI, including the global optimal solution, PI tube movement, asymptotic guarantees, and distribution-free estimation, while also achieving a sparse solution.

---
**Algorithm 2** PI estimation through SSVQR
---
1: **procedure** :- PI THROUGH SSVQR$(T, 1 - \alpha)$
2:    Choose some $\bar{q} \in (0, 1 - \alpha)$
3:    **for**  each $q \in \{\bar{q}, (1 + \bar{q} - \alpha)\}$ **do**
4:       Solve the LPP problem (18) by tuning the value of $C$. Obtain the solution $(r^*, p^*, b^*)$.
5:       Estimate the function $f_q(x)$ using the (19).
6:    **return** $(f_{\bar{q}}(x), f_{1+\bar{q}-\alpha}(x))$

---

## 4.3   Feature selection in PI estimation through SSVQR

Similar to other machine learning tasks, the PI estimation in high-dimensional settings also presents several challenges.  The increased dimensionality not only increases the complexity of the PI bounds but also necessitates a larger sample size to ensure the quality of the estimate PI. Therefore, an efficient feature selection method is crucial for reducing the overall complexity of the PI estimation, particularly when dealing with high-dimensional data.

We detail the feature selection algorithm through SSVQR model for the linear PI estimation task in Algorithm 3.  Here, a linear PI refers to the PI, where both bounds are linear functions of the input variables ,i,e. $f_{\bar{q}}(x) = w_{\bar{q}}^T x + b_{\bar{q}} \ f_{1+\bar{q}-\alpha}(x) = w_{1+\bar{q}-\alpha}^T x + b_{1+\bar{q}-\alpha}.$

The input of Algorithm 3 is the training set $T' = \{(x_i, y_i), x_i \in \mathbf{R}^n, y_i \in \mathbf{R}, i = 1, 2, ....m\}$, specified confidence $(1-\alpha)$, and a very small number $\epsilon$. It finally returns the selected feature set. In the next section, we have explored several real-world datasets with numerous features and successfully perform feature selection using the SSVQR method for linear PI estimation, without compromising the quality of the estimated PI.

---

**Algorithm 3** Feature selection through SSVQR

---
1: **procedure** :- FEATURE SELECTION THROUGH SSVQR$(T, 1 - \alpha, \epsilon)$
2:     Choose some $\bar{q} \in (0, 1 - \alpha)$
3:     **for** each $q \in \{\bar{q}, (1 + \bar{q} - \alpha)\}$ **do**
4:         Consider the linear kernel $k(x_i, x_j) = x_i^T x_j$ at the LPP (18).
5:         Solve the LPP (18) and obtain its solution $(r^*, p^*, b^*)$.
6:         Obtain the $w_q$ using $[x_1, x_2, ..., x_m](r^* - p^*)$.
7:     Compute $I_{\bar{q}} = \{i :, |w_{\bar{q}}(i)| \leq \epsilon\}$ and $I_{1+\bar{q}-\alpha} = \{i :, |w_{1+\bar{q}-\alpha}(i)| \leq \epsilon\}$
8:     Compute $I = I_{\bar{q}} \cap I_{1+\bar{q}-\alpha}$ and *Feature Set* $= \{1, 2, ..., m\} - I$
9:     **return** (*Feature Set*)

---

### 4.4 Conformal Regression in SVM

Finally, we extend the SVM models in CR setting for obtaining the finite sample test set guarantees. In split CR setting, we detail the SVM based CR algorithm as follows.

---

**Algorithm 4** CR through SVQR

---
1: **procedure** :- CR THROUGH SVQR$(T, 1 - \alpha)$
2:     Split the training set $T'$ into $I_1$ and calibration set $I_2$.
3:     Choose some $\bar{q} \in (0, 1 - \alpha)$
4:     **for** each $q \in \{\bar{q}, (1 + \bar{q} - \alpha)\}$ **do**
5:         Solve the QPP problem (4) on $I_1$ by tuning the value of $C$. Obtain the solution $(\alpha^*, \beta^*)$.
6:         Estimate the function $f_q(x)$ using the (5).
7:     Evaluate the nonconformity score $E_i = \max\{f_{\bar{q}}(x) - y_i, \ y_i - f_{1-\bar{q}+\alpha}(x)\}$ on $I_2$.
8:     Compute the $Q_{1-\alpha}(E, I_2) = (1 - \alpha)(1 + \frac{1}{|I_2|})$-th empirical quantile of the non-conformity score $E_i$.
9:     **return** $C(x_{m+1}) = [f_{\bar{q}}(x_{m+1}) - Q_{(1-\alpha)}(E, I_2), f_{1-\bar{q}+\alpha}(x_{m+1}) + Q_{1-\alpha}(E, I_2)]$

---

Compared to neural network (NN)-based CR models, the SVM-based CR model offers not only greater interpretability but also more stable prediction sets. In contrast, NN-based CR models often produce varying prediction set across different training runs, even when trained on the same data splits ($I_1$ and $I_2$) and with identical hyperparameter settings. It is because that unlike SVMs, NN models often converge to different local optima during training. We empirically verify these advantages of the SVM based CR model in next section.

## 5 Experimental Results

In this section, we present the numerical results to analyze the quality of the PI obtained by the different SVM models through a series of experiments on simulated/artificial and benchmark datasets. But, before this, we outline the objective of our experiments in view of our claim as follows.

(i) We need to evaluate the effectiveness of the SSVQR model and assess the quality of its PI estimate relative to other existing PI methods in SVM on simulated and benchmark datasets.

(ii) We aim to validate the effectiveness of Algorithm 3 for feature selection by demonstrating its ability to identify relevant features for PI estimation. Furthermore, we check that whether it improves the quality of PIs, particularly in high-dimensional real-world benchmark datasets under the linear PI estimation setting.

(iii) We need to compare the SVM based CR and NN based CR model to verify our claim that prediction sets obtained by SVM based CR model are more stable and less uncertain than that of NN.

(iv) We need to check the potential of SVM models for probabilistic foretasting tasks against the recently developed deep learning solutions.

We have made the code and datasets used in our experiments available at: `https://github.com/anonymousmyaccount/myaccount/tree/main/-PI-IN-SVM--my_code` while ensuring that anonymity is preserved.

**Evaluation metrics and criteria**

Now, we describe in detail the evaluation criteria that will be used for our experiments. In all our experiments, the objective is to estimate PI with a confidence level of either $1 - \alpha = 0.95$ or $1 - \alpha = 0.90$. To construct a PI with $1 - \alpha$ confidence level, we seek the pair of quantile functions $(f_q(x), f_{1+q-\alpha}(x))$, where $0 \leq q \leq 0.05$.

In case of artificial datasets, we know the noise distribution and the true quantile function can be easily computed by the inverse of the cumulative distribution function. Therefore, the quality of the quantile function can be accurately assessed by computing the RMSE between the true and estimated quantile functions.

However, to estimate the quality of the quantile function in the absence of information about the noise distribution, we can use the Coverage Probability (CP). For given a test set $\{(x_i, y_i) : x_i \in \mathbf{R}^n, y \in \mathbf{R}, i = 1, 2, ....k\}$ and estimate of the $q^{th}$ quantile function $f_q(x)$, the CP is computed by $\frac{|\{y_i : y_i \leq f_q(x_i)\}|}{k}$. It measures the fraction of $y$ values falling below the estimated quantile function.

For evaluating the overall quality of PI estimation, we use PICP and MPIW as assessment criteria. If the $[f_q(x), f_{1+q-\alpha}(x)]$ is the estimate of the PI, then the PICP on test set is computed by $\frac{1}{k}\#\{y_i : f_q(x_i) \leq y_i \leq f_{1+q-\alpha}(x_i), i = 1, 2, ..k.\}$, where $\#$ denotes the cardinality of the set. Also, the MPIW can be computed by $\frac{1}{k}\sum_{i=1}^{k}|f_{1+q-\alpha}(x_i)\} - f_q(x_i)|$.

An effective PI method must achieve the target $1 - \alpha$ calibration while minimizing the PI width, as measured by the MPIW value. Furthermore, we define the Prediction Interval Coverage Error (PICE) as $\max(0, (1 - \alpha) - PICP)$ to quantify the extent to which the model falls short of the target calibration (1-$\alpha$). For comparing PI estimation models, a natural decision criterion is that the model with the lowest PICE should be considered the best. If all models successfully achieve the target calibration, the one with the minimum MPIW should be deemed the most optimal.

## 5.1 PI estimation through SSVQR model

At first, we shall perform experiments with the SSVQR PI estimation model on artificial and benchmark datasets.

**Baseline methods**

To evaluate the quality of the PIs produced by the SSVQR model, it is essential to establish appropriate baseline methods for comparison. One of the key strengths of SVM machines is their ability to always obtain the global optimal solution, maintaining their relevance and applicability in modern cutting-edge technology. As detailed in Table 1, PI estimation through SVQR and LS-SVR only obtains the global optimal solution but lacks sparsity. In contrast, the SSVQR PI can obtain the optimal global solution as well as the sparse solution. In view of this, we find the SVQR and LS-SVR models are qualified enough to be compared with the SSVQR model for the PI estimation task in SVM framework. The Tube and LUBE loss PI estimation methods available in SVM framework do not guarantee the global optimal solution and their solution may vary with the choice of initialization. However, we have considered the Tube loss and an improved version of the LUBE loss, the QD loss function (Pearce et al. (2018)) in deep forecasting models to compare their performance with the SVM based probabilistic forecasting models.

**Experimental Setup and Parameter Tunning**

All SVM based PI estimation models have been implemented in MATLAB. To estimate both bounds of the PI, we utilize the RBF kernel, defined as $k(x_i, x_j) = e^{-\gamma||x_i - x_j||^2}$. As detailed in Algorithm 1 and Algorithm 2, SVQR and SSVQR require solving the QPP (4) and LPP (18) twice to obtain the quantile bounds of the

| | $(\bar{q},\ 1+\bar{q}-\alpha)$ | RMSE(Lw,Up) | Spar (Lw,Up) | CP (Lw,Up) | PICP | PICE | MPIW | Time (s) |
|---|---|---|---|---|---|---|---|---|
| | (0.01, 0.96) | (1.8021, 1.4446) | (0%, 0%) | (0.0110, 0.9680) | 0.957 | 0 | 2.9054 | 0.6157 |
| | (0.015, 0.965) | (1.7046, 1.4653) | (0%, 0%) | (0.0150, 0.9720) | 0.957 | 0 | 2.8244 | 0.5844 |
| SVQR | (0.02, 0.97) | (1.6519, 1.4799) | (0%, 0%) | (0.0180, 0.9720) | 0.954 | 0 | 2.7855 | 0.6169 |
| | (0.025, 0.975) | (1.5857, 1.5016) | (0%, 0%) | (0.0220, 0.9760) | **0.954** | 0 | **2.7379** | 0.5546 |
| | (0.03, 0.98) | **(1.5030, 1.5667)** | (0%, 0%) | (0.0250, 0.9800) | 0.955 | 0 | 2.7212 | **0.5150** |
| | (0.01, 0.96) | (1.8060, 1.4418) | (15%, 18%) | (0.0110, 0.9680) | 0.957 | 0 | 2.9056 | 0.4783 |
| | (0.015, 0.965) | (1.7051, 1.4714) | (15%, 18%) | (0.0150, 0.9700) | 0.955 | 0 | 2.8275 | 0.5485 |
| SSVQR | (0.02, 0.97) | (1.6497, 1.4861) | (15%, 18%) | 0.0160, 0.9720 | 0.956 | 0 | 2.7939 | **0.4712** |
| | (0.025, 0.975) | **(1.5517, 1.4978)** | (15%, 20%) | (0.0250, 0.9750) | **0.950** | 0 | **2.6995** | 0.4928 |
| | (0.03, 0.98) | (1.5133, 1.5604) | **(16%, 20%)** | (0.0290, 0.9800) | 0.951 | 0 | 2.7288 | 0.6054 |

Table 2: Performance of the SVQR and SSVQR PI models on AD 1 dataset. Lw: Lower, Up: Upper, Spar: Sparsity, Time(s) : Training time in seconds. SSVQR PI models always obtain the sparse solution. Also, for $\bar{q} = 0.025$, SSVQR model obtains the best quality PI with MPIW = 2.70 and PICP = 0.95.

PI respectively. We solve the QPPs of the SVQR PI model and the LPPs of the SSVQR model in MATLAB using the 'quadprog' and 'linprog' packages respectively.

The 'linprog' function employs the "dual-simplex-highs" algorithm by default, which may not scale efficiently for very large datasets. A more suitable open-source alternative for solving the LPPs in the SSVQR PI model is the "HiGHS" solver, which is specifically designed to handle large-scale linear programming problems involving thousands or even millions of variables and constraints with high efficiency.

The SVQR problem(4) or SSVQR (18) problem requires the supply of the two user defined parameters $C$ and RBF kernel parameter $\gamma$ for non-linear PI estimation. We have tunned the value of the these parameters using the grid search in the search space $\{2^{-8}, 2^{-7}, ...., 2^7, 2^8\} \times \{2^{-8}, 2^{-7}, ..., 2^7, 2^8\}$.

**Artificial Datasets**

First, we generate six distinct artificial datasets. In each dataset, the values of $x_i$ are randomly sampled from a uniform distribution $U(-5, 5)$, while the corresponding $y_i$ values are obtained by polluting the function $(1 - x_i + 2x_i^2)e^{-0.5x_i^2}$ with different types of noise as described below.

$$\textbf{AD 1:-}\ \ y_i = (1 - x_i + 2x_i^2)e^{-0.5x_i^2} + \xi_i, \text{where}\ \ \xi_i \sim N(0, 0.6)$$

$$\textbf{AD 2:-}\ \ y_i = (1 - x_i + 2x_i^2)e^{-0.5x_i^2} + \xi_i, \quad \text{where}\ \ \xi_i \sim \chi^2(3)$$

$$\textbf{AD 3:-}\ \ y_i = (1 - x_i + 2x_i^2)e^{-0.5x_i^2} + \xi_i, \text{where}\ \ \xi_i \sim N(0, 0.4)$$

$$\textbf{AD 4:-}\ \ y_i = (1 - x_i + 2x_i^2)e^{-0.5x_i^2} + \xi_i, \text{where}\ \ \xi_i \sim N(0, 0.8)$$

$$\textbf{AD 5:-}\ \ y_i = (1 - x_i + 2x_i^2)e^{-0.5x_i^2} + \xi_i, \text{where}\ \ \xi_i \sim U(-5, 5)$$

$$\textbf{AD 6:-}\ \ y_i = (1 - x_i + 2x_i^2)e^{-0.5x_i^2} + \xi_i, \text{where}\ \ \xi_i \sim U(-4, 4)$$

In case of each dataset, 2500 data points are generated in which 1000 data points are considered for training and rest of them are considered for testing.

**Artificial Datasets Results, Discussion and Analysis**

We present the performance of the SVQR and SSVQR model for PI estimation task with the different values of $\bar{q}$ for each of simulated dataset in Table 2-7. The leftmost column of these Tables list the different pairs of target quantiles $(\bar{q},\ 0.95 + \bar{q})$, required for the PI estimation. As detailed in Section 5.1 of this paper, for artificial datasets, the quality of the estimated upper and lower quantiles of the PI can be best evaluated by computing the RMSE between the estimated quantiles and their corresponding true quantile functions. The third column of the Tables 2-7 list these RMSE for different values of $\bar{q}$.

To effectively illustrate the comparative performance between the SVQR and SSVQR models across various artificial datasets, we normalize the MPIW values by dividing them by the variance of each dataset (AD1– AD6). Additionally, we compute the aggregated RMSE as the mean of RMSE (Lp) and RMSE (Up), which represent the RMSE values corresponding to the lower and upper quantile bounds of the PI, respectively.

| | $(\bar{q},\ 1+\bar{q}-\alpha)$ | RMSE(Lw,Up) | Spar (Lw,Up) | CP (Lw,Up) | PICP | PICE | MPIW | Time (s) |
|---|---|---|---|---|---|---|---|---|
| | (0.01, 0.96) | **(4.2653, 5.7073)** | (0%, 0%) | (0.0100, 0.9470) | 0.937 | 0.013 | 8.3755 | 0.805 |
| | (0.015, 0.965) | (4.1993, 6.0698) | (0%, 0%) | (0.0210, 0.9520) | 0.931 | 0.019 | 8.6865 | 0.699 |
| SVQR | (0.02, 0.97) | (4.1312, 6.5751) | (0%, 0%) | (0.0220, 0.9610) | **0.939** | **0.011** | 9.1619 | 0.6595 |
| | (0.025, 0.975) | (4.0525, 6.8163) | (0%, 0%) | (0.0340, 0.9650) | 0.931 | 0.019 | 9.3256 | 0.719 |
| | (0.03, 0.98) | (4.0251, 7.4745) | (0%, 0%) | (0.0350, 0.9720) | 0.937 | 0.013 | 9.9780 | 0.6809 |
| | (0.01, 0.96) | **(4.1628, 5.5481)** | (20%, 15%) | (0.0080, 0.9390) | 0.931 | 0.019 | 8.1036 | **0.502** |
| | (0.015, 0.965) | (4.0206, 5.9335) | (18%, 18%) | (0.0100, 0.9520) | 0.942 | 0.008 | 8.3460 | 0.5173 |
| SSVQR | (0.02, 0.97) | (4.0156, 6.3077) | **(20%, 25%)** | (0.0110, 0.9640) | **0.953** | **0** | **8.7801** | 0.5128 |
| | (0.025, 0.975) | (3.9444, 7.1596) | (15%, 20%) | (0.0240, 0.9680) | 0.944 | 0.006 | 9.5723 | 0.4694 |
| | (0.03, 0.98) | (3.9309, 7.3834) | (20%, 20%) | (0.0290, 0.9740) | 0.945 | 0.005 | 9.8075 | 0.5113 |

Table 3: Performance of the SVQR and SSVQR PI models on AD 2 dataset. Lw: Lower, Up: Upper, Spar: Sparsity, Time(s) : Training time in seconds. SSVQR PI models always obtain the sparse solution. Also, for $\bar{q} = 0.02$, SSVQR model obtains the best quality PI with PCIP = 0.953 and MPIW = 8.78.

| | $(\bar{q},\ 1+\bar{q}-\alpha)$ | RMSE(Lw,Up) | Spar (Lw,Up) | CP (Lw,Up) | PICP | PICE | MPIW | Time (s) |
|---|---|---|---|---|---|---|---|---|
| | (0.01, 0.96) | (1.7894, 1.5837) | (0%, 0%) | ( 0.0120, 0.9570) | 0.945 | 0.050 | 2.9000 | 0.4872 |
| | (0.015, 0.965) | (1.7355, 1.6297) | (0%, 0%) | (0.0130, 0.9590) | **0.946** | 0.004 | **2.8870** | 0.5077 |
| SVQR | (0.02, 0.97) | **(1.5696, 1.6407)** | (0%, 0%) | (0.0230, 0.9590) | 0.936 | 0.014 | 2.7223 | 0.5004 |
| | (0.025, 0.975) | (1.5070, 1.6927) | (0%, 0%) | (0.0280, 0.9680) | 0.940 | 0.010 | 2.7100 | 0.4907 |
| | (0.03, 0.98) | (1.4694, 1.7477) | (0%, 0%) | (0.0300, 0.9750) | 0.945 | 0.005 | 2.7394 | **0.4745** |
| | (0.01, 0.96) | 1.8411, 1.5781 | (20%, 40%) | 0.0120, 0.9570 | 0.945 | 0.005 | 2.9462 | **0.4367** |
| | (0.015, 0.965) | (1.8297, 1.5917) | (40%, 30%) | (0.0120, 0.9570) | **0.945** | 0.005 | **2.9458** | 0.4563 |
| SSVQR | (0.02, 0.97) | (1.6999, 1.6969) | (40%, 40%) | (0.0180, 0.9600) | 0.942 | 0.008 | 2.9026 | 0.4671 |
| | (0.025, 0.975) | (1.6509, 1.7191) | **(40%, 40%)** | (0.0220, 0.9620) | 0.940 | 0.010 | 2.8817 | 0.4501 |
| | (0.03, 0.98) | **(1.4865, 1.7721)** | (30% , 40%) | (0.0290, 0.9700) | 0.941 | 0.009 | 2.7739 | 0.4717 |

Table 4: Performance of the SVQR and SSVQR PI models on AD 3 dataset. Lw: Lower, Up: Upper, Spar: Sparsity, Time(s) : Training time in seconds. SSVQR PI models always obtain the sparse solution. SSVQR and SVQR model tends to obtain the similar performance.

| | $(\bar{q},\ 1+\bar{q}-\alpha)$ | RMSE(Lw,Up) | Spar (Lw,Up) | CP (Lw,Up) | PICP | PICE | MPIW | Time (s) |
|---|---|---|---|---|---|---|---|---|
| | (0.01, 0.96) | (2.5953, 2.1197) | (0%, 0% ) | (0.0160, 0.9530) | 0.937 | 0.013 | 4.1549 | 0.723 |
| | (0.015, 0.965) | (2.5595, 2.1292) | (0%, 0%) | (0.0160, 0.9530) | 0.937 | 0.013 | 4.1268 | 0.6713 |
| SVQR | (0.02, 0.97) | (2.4221, 2.2193) | (0%, 0%) | (0.0200, 0.9610) | **0.941** | 0.009 | **4.0817** | 0.6704 |
| | (0.025, 0.975) | **(2.2561, 2.2555)** | (0%, 0%) | (0.0290, 0.9630) | 0.934 | 0.016 | 3.9365 | 0.6815 |
| | (0.03, 0.98) | (2.2211, 2.3699) | (0%, 0%) | (0.0310, 0.9670) | 0.936 | 0.014 | 4.0273 | **0.6343** |
| | (0.01, 0.96) | (2.5842, 2.1364) | (20%, 40%) | (0.0160, 0.9530) | 0.937 | 0.013 | 4.1577 | 0.4704 |
| | (0.015, 0.965) | (2.5475, 2.1911) | (20%, 20%) | (0.0160, 0.9560) | 0.940 | 0.010 | 4.1802 | 0.4745 |
| SSVQR | (0.02, 0.97) | (2.4939, 2.2455) | (20%, 20%) | (0.0190, 0.9610) | **0.942** | 0.008 | **4.1830** | 0.4818 |
| | (0.025, 0.975) | **(2.3617, 2.3037)** | (40%, 20%) | (0.0240, 0.9640) | 0.940 | 0.010 | 4.1002 | **0.4662** |
| | (0.03, 0.98) | (2.2689, 2.4147) | (40%, 20%) | (0.0290, 0.9700) | 0.941 | 0.009 | 4.1210 | 0.4884 |

Table 5: Performance of the SVQR and SSVQR PI models on AD 4 dataset. Lw: Lower, Up: Upper, Spar: Sparsity, Time(s) : Training time in seconds. SSVQR PI models always obtain the sparse solution. SSVQR and SVQR model tends to obtain the similar performance.

| | $(\bar{q},\ 1+\bar{q}-\alpha)$ | RMSE(Lw,Up) | Spar (Lw,Up) | CP (Lw,Up) | PICP | PICE | MPIW | Time (s) |
|---|---|---|---|---|---|---|---|---|
| | (0.01, 0.96) | (4.9694, 4.4016) | (0%, 0% ) | **(0.0100, 0.9560)** | **0.946** | 0.004 | **8.1044** | 0.7917 |
| | (0.015, 0.965) | (4.8456, 4.4539) | (0%, 0%) | (0.0180, 0.9600) | 0.942 | 0.008 | 8.0257 | 0.8009 |
| SVQR | (0.02, 0.97) | (4.8100, 4.4860) | (0%, 0%) | (0.0190, 0.9620) | 0.943 | 0.007 | 8.0223 | 0.736 |
| | (0.025, 0.975) | **(4.7111, 4.5143)** | (0%, 0%) | (0.0280, 0.9640) | 0.936 | 0.014 | 7.9430 | 0.6895 |
| | (0.03, 0.98) | (4.6921, 4.5368) | (0%, 0%) | (0.0290, 0.9680) | 0.939 | 0.011 | 7.9481 | **0.6191** |
| | (0.01, 0.96) | (5.2658, 4.5044) | (40%, 30%) | **(0.0050, 0.9610)** | **0.956** | 0.00 | **8.5344** | 0.5394 |
| | (0.015, 0.965) | (4.8787, 4.5073) | (35%, 30%) | (0.0160, 0.9610) | 0.945 | 0.005 | 8.1171 | 0.4835 |
| SSVQR | (0.02, 0.97) | **(4.8138, 4.5249)** | (30%, 30%) | (0.0190, 0.9650) | 0.946 | 0.004 | 8.0659 | **0.472** |
| | (0.025, 0.975) | (4.7464, 4.6037) | (25%, 30%) | (0.0260, 0.9670) | 0.941 | 0.009 | 8.0692 | 0.5102 |
| | (0.03, 0.98) | (4.6962, 4.7637) | **(30%, 40%)** | (0.0290, 0.9720) | 0.943 | 0.007 | 8.1832 | 0.5068 |

Table 6: Performance of the SVQR and SSVQR PI models on AD 5 dataset. Lw: Lower, Up: Upper, Spar: Sparsity, Time(s) : Training time in seconds. SSVQR PI models always obtain the sparse solution. For $\bar{q} = 0.01$, SSVQR PI model obtains the best quality PI with PICP =0.956 and MPIW = 8.53.

|  | $(\bar{q},\ 1+\bar{q}-\alpha)$ | RMSE(Lw,Up) | Spar (Lw,Up) | CP (Lw,Up) | PICP | PICE | MPIW | Time (s) |
|---|---|---|---|---|---|---|---|---|
|  | **(0.01, 0.96)** | (6.1510, 5.5057) | (0%, 0%) | **(0.0110, 0.9630)** | **0.952** | 0 | **10.0826** | **0.6437** |
|  | (0.015, 0.965) | (6.0517, 5.5584) | (0%, 0%) | (0.0150, 0.9670) | 0.952 | 0 | 10.031 | 0.6798 |
| SVQR | (0.02, 0.97) | (5.9777, 5.6130) | (0%, 0% ) | (0.0170, 0.9700) | 0.953 | 0 | 10.0141 | 1.0752 |
|  | (0.025, 0.975) | (5.8879, 5.6868) | (0%, 0%) | (0.0230, 0.9760) | 0.953 | 0 | 9.9987 | 0.9265 |
|  | (0.03, 0.98) | **(5.7599, 5.7156)** | (0%, 0%) | (0.0290, 0.9780) | 0.949 | 0.001 | 9.8878 | 0.9445 |
|  | (0.01, 0.96) | (5.9304, 5.4581) | (10%, 15%) | (0.0110, 0.9610) | 0.95 | 0 | 9.7944 | 0.569 |
|  | (0.015, 0.965) | (5.9107, 5.5686) | (10%, 20%) | (0.0140, 0.9700) | 0.956 | 0 | 9.9016 | 0.5656 |
| SSVQR | **(0.02, 0.97)** | (5.7836, 5.5880) | (15%, 25%) | **(0.0180, 0.9700)** | **0.952** | 0 | **9.7807** | 0.5766 |
|  | (0.025, 0.975) | **(5.6486, 5.6038)** | **(20%, 20%)** | (0.0260, 0.9710) | 0.945 | 0.005 | 9.6443 | **0.5423** |
|  | (0.03, 0.98) | (5.6184, 5.6506) | (10%, 15%) | (0.0300, 0.9760) | 0.946 | 0.004 | 9.6653 | 0.5636 |

Table 7: Performance of the SVQR and SSVQR PI models on AD 6 dataset. Lw: Lower, Up: Upper, Spar: Sparsity, Time(s):Training time in seconds. SSVQR PI models always obtain the sparse solution. For $\bar{q} = 0.01$, SSVQR PI model obtains the best quality PI with PICP =0.956 and MPIW = 8.53.

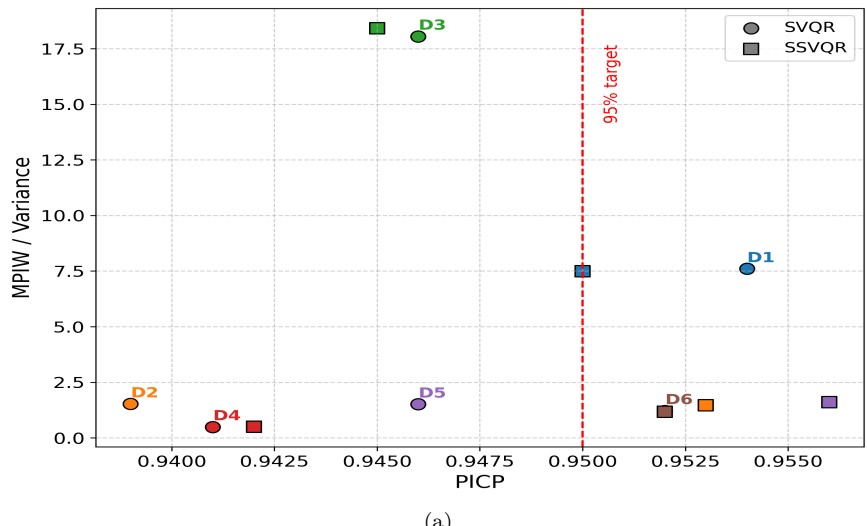

(a)

Figure 3: Comparison of the quality of PIs estimated by the SVQR and SSVQR models. For a high-quality PI, the PICP should be greater than or equal to the target value of $(1-\alpha) = 0.95$, while maintaining the smallest possible MPIW values. Among the six datasets (AD1–AD6), the SSVQR-based PI model successfully achieves the target PICP of 0.95 on four datasets, whereas the SVQR-based PI model fails to meet the target PICP on four datasets.

Further, we also normalize them by dividing the variance present in each datasets for the comparative study.

Figure 3 shows the comparison of the quality of PIs estimated by the SVQR and SSVQR models. For a high-quality PI, the PICP should be greater than or equal to the target value of $(1-\alpha) = 0.95$, while maintaining the smallest possible MPIW values. Among the six datasets (AD1-AD6), the SSVQR-based PI model successfully achieves the target PICP of 0.95 on four datasets, where as the SVQR-based PI model fails to meet the target PICP on four datasets. Figure 4 illustrates the comparisons of the normalized aggregated RMSE obtained by the SVQR and SSVQR models. The quantile bounds estimated by the SSVQR model are comparable to, or slightly better than, those obtained from the SVQR model.

Replacing $L_2$ regularization with $L_1$ regularization in the SVM quantile regression model results in improved PI quality. However, the major advantage of using the SSVQR model over SVQR model is its ability to obtain the sparse solution vector. Figure 5(a) plots the sparsity of the solution vector corresponding to the upper and lower quantile bounds obtained by the SSVQR model. It highlights that the SSVQR model effectively reduces significant coefficients of the solution vector to near zero which enables the feature selection task in PI estimation and also simplify the overall prediction process.

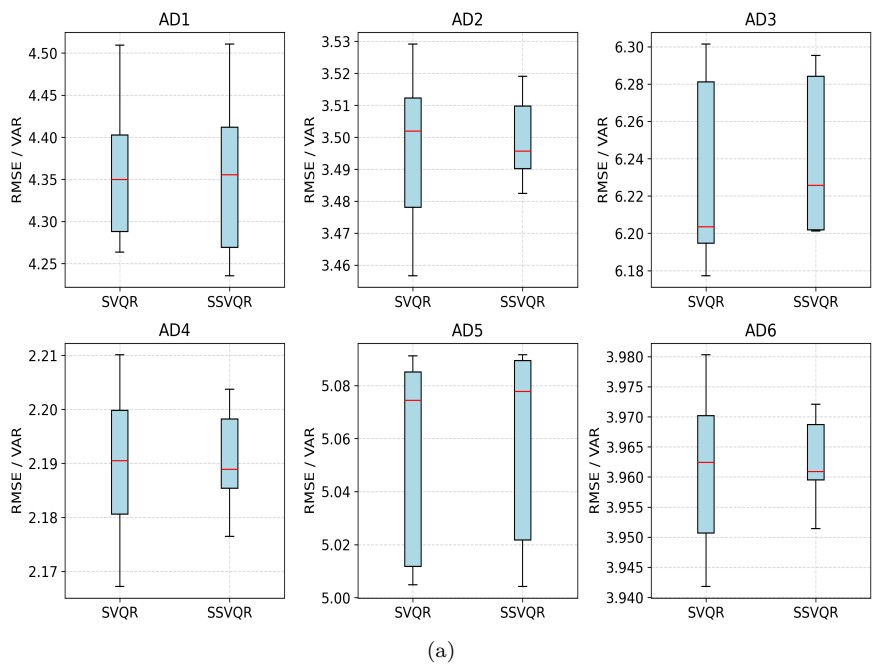

(a)

Figure 4: Comparison of the quality of the quantile functions estimated by the SVQR and SSVQR models is performed using the aggregated RMSE metric. The aggregated RMSE is computed as the mean of RMSE(Lw) and RMSE(up) which represent the RMSE values for the lower and upper quantile bounds of the PI, respectively. These RMSE values are then normalized by dividing by the variance of each dataset.

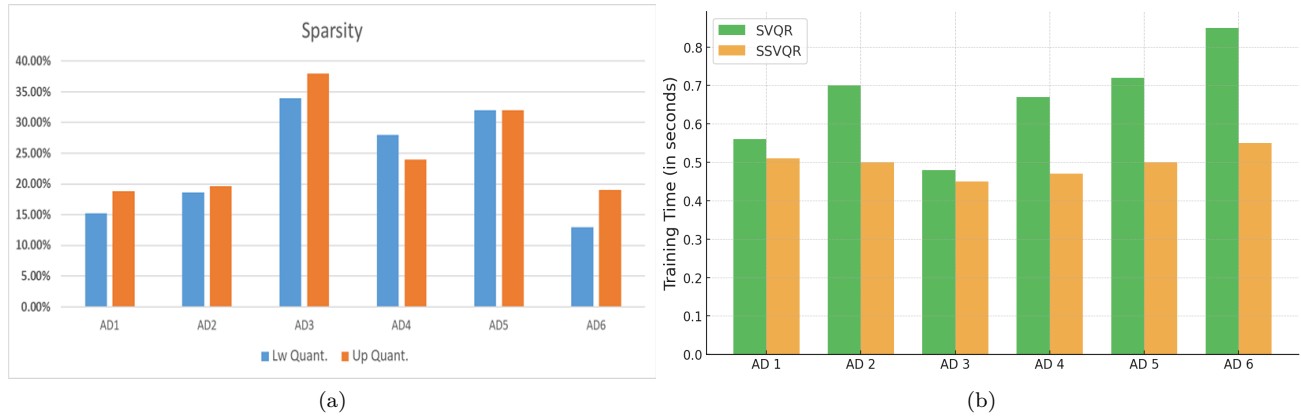

(a)                                                                          (b)

Figure 5: (a) Qunatile functions estimated by the SSVQR model is sparse while SVQR model fails to obtain the sparse solution. (b) Average training time comparison of the SVQR and SSVQR models for PI estimation.

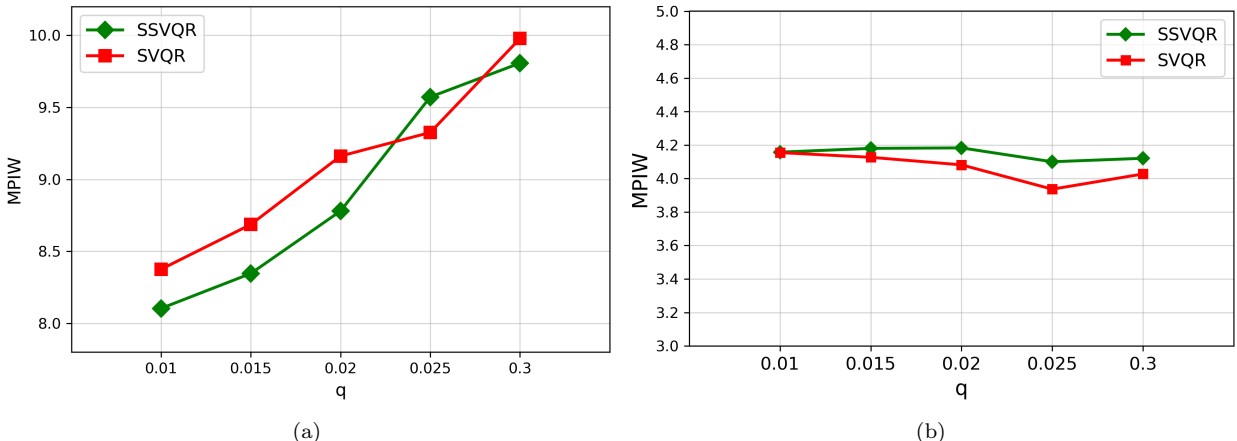

(a)                                                          (b)

Figure 6: Plot of the MPIW values obtained by the SVQR and SSVQR PI models as a function of $\bar{q}$ for (a) the AD2 dataset and (b) the AD4 dataset. The AD2 dataset contains asymmetric noise drawn from a $\chi^2$ distribution, resulting in a higher concentration of data points in the lower region of the input–target space. Consequently, the MPIW increases as the PI tube shifts upward with larger values of $\bar{q}$. In contrast, the AD4 dataset contains noise from a normal distribution; therefore, the centered PI obtained with the $\bar{q} = 0.025$ yields the narrower PI.

We compare the overall average training time (in seconds) taken by the SVQR PI and SSVQR PI models for the PI estimation task across different artificial datasets in Figure 5(b). It shows that the SSVQR requires fewer seconds train the PI model than the SVQR model. We have solved the QPPs of the SVQR PI model and LPPs of the SSVQR model in MATLAB with 'quadprog' and 'linprog' packages respectively.

Another key observation from the numerical results in Tables 2-7 is the consistent performance of the SVQR and SSVQR PI models across different datasets. In all cases, both the SVQR and SSVQR PI models manage to approximately achieve 95% target coverage. We plot the MPIW values obtained by the SSVQR and SVQR PI models against different values of $\bar{q}$ on dataset AD2 and AD4 in Figure 6. On AD2 dataset, the MPIW values of the PI obtained by both models increase with $\bar{q}$. It is evident from Table 3 that this increase is not related to the PICP values obtained by the SVQR and SSVQR PI models. Actually, it is caused by the nature of noise present in the AD2 dataset. The AD2 dataset contains asymmetric noise from the $(\chi^2)$ distribution, leading to a higher density of data points in the lower region of the input-target space, which gradually decreases as we move upward. As $\bar{q}$ decreases, the resultant PI shifts downward, passing through denser regions of the data cloud, leading to lower MPIW values. It shows that the movement of the PI tube due to change of $\bar{q}$ values may lead to the narrower PI particularly in presence of the asymmetric noise in the data. Apart from the AD2 dataset, all other artificial datasets contain noise from symmetric distributions. In these datasets, the centered PI $((f_{0.025}(x), f_{0.975}(x))$ is expected to achieve the high quality PI. Figure 6(b) shows that at $\bar{q} = 0.025$, both the SVQR and SSVQR PI models attain the lowest MPIW values on AD4 dataset.

**SVM PI estimation methods on benchmark datasets**

We have done experiments on the two popular benchmark datasets namely Boston Housing (Harrison Jr & Rubinfeld (1978)) and Concrete (Yeh (1998)) and evaluate the quality of the PI estimated by the SSVQR, SVQR and LS-SVR based PI estimation methods for different value of the $\bar{q}$ with non-linear RBF kernel. Table 8 and 9 contains the numerical results obtained on the Boston Housing and Concrete datasets respectively. We can observe that the SSVQR, SVQR and LS-SVR PI models obtains a similar quality of the PI but, SSVQR models always obtain the sparse solution vector.

| | q | Spar (Lw, Up) | CP (Lw, Up) | PICP | PICE | MPIW | Time (s) |
|---|---|---|---|---|---|---|---|
| | (0.025, 0.925) | (0%, 0%) | (0.028, 0.930) | **0.90** | 0 | **28.55** | **0.1347** |
| SVQR | (0.05, 0.95) | (0%, 0%) | (0.052, 0.950) | 0.90 | 0.00 | 33.06 | 0.1472 |
| | (0.075, 0.975) | (0%, 0%) | (0.088, 0.972) | 0.88 | 0.02 | 38.23 | 0.1670 |
| | **(0.025, 0.925)** | **(17%, 16%)** | (0.027, 0.927) | **0.90** | 0 | 28.62 | **0.0657** |
| SSVQR | (0.05, 0.95) | (15%, 22%) | (0.048, 0.947) | 0.90 | 0 | 32.77 | 0.0678 |
| | (0.075, 0.975) | (14%, 28%) | (0.080, 0.967) | 0.89 | 0.01 | 38.24 | 0.0681 |
| | (0.025, 0.925) | (0%,0% ) | | 0.91 | 0 | 31.19 | **0.0064** |
| LS-SVR | **(0.050, 0.950)** | (0%,0%) | | **0.91** | 0 | **30.18** | 0.0067 |
| | (0.075, 0.975) | (0%,0%) | | 0.89 | 0.01 | 31.19 | 0.0060 |

Table 8: Comparison of different SVM PI estimation methods on Boston Housing. Lw: Lower, Up: Upper, Spar: Sparsity, Time(s) : Training time in seconds.

| | q | Spar (Lw, Up) | CP (Lw, Up) | PICP | PICE | MPIW | Time (s) |
|---|---|---|---|---|---|---|---|
| | (0.025, 0.925) | (0%, 0%) | (0.0219, 0.8997) | 0.8777 | 0.0723 | 32.5492 | 2.6141 |
| SVQR | **(0.05, 0.95)** | (0%, 0%) | (0.0408, 0.9436) | **0.9028** | **0.0472** | **28.6541** | 2.8405 |
| | (0.075, 0.975) | (0%, 0%) | (0.0690, 0.9561) | 0.8871 | 0.0629 | 32.1635 | **2.5197** |
| | (0.025, 0.925) | (12%, 12%) | (0.0340, 0.9029) | 0.8689 | 0.0811 | 29.4962 | 0.1858 |
| SSVQR | (0.05, 0.95) | (12%, 12%) | (0.0583, 0.9272) | 0.8689 | 0.0811 | **27.8917** | **0.1795** |
| | **(0.075, 0.975)** | **12%, 12%** | (0.0777, 0.9515) | **0.8738** | 0.0762 | 30.2547 | 0.1837 |
| | (0.025, 0.925) | 0%,0% | | 0.9126 | 0.0374 | 28.2429 | **0.4125** |
| LS-SVR | (0.050, 0.950) | (0%,0%) | | 0.8932 | 0.0568 | **27.3308** | 0.4631 |
| | **(0.075, 0.975)** | (0%,0%) | | **0.9029** | 0.0471 | 28.2429 | 0.3954 |

Table 9: Comparison of different SVM PI estimation methods on Concrete dataset.Lw: Lower, Up: Upper, Spar: Sparsity, Time(s) : Training time in seconds.

## 5.2 Feature Selection through SSVQR

Next, we apply our SSVQR model for feature selection in PI estimation with linear kernel.

Experimental Setup:- All of the simulations were performed in the MATLAB. We use the 80% of data points for training and use rest of them for testing. We set the target calibration $1-\alpha = 0.95$. Following Algorithm 3, we perform feature selection using the SSVQR PI model for $\bar{q} = 0.025$ and present the results in Table 10. Since Algorithm 3 solves the problem 18 with linear kernel twice, we need to tune only the parameter $C$ for each $\bar{q}$ which was selected from the grid of $\{2^i : i = -15, ...., 15\}$.

Datasets:- we use five popular real-world benchmark datasets namely Spambase ($4601 \times 58$) (Hopkins & Suermondt (1999)), Student Performance ($395 \times 16$) (Cortez (2008)) , Boston Housing ($505 \times 14$) (Harrison Jr & Rubinfeld (1978)), UCI-secom ($1568 \times 591$) (McCann & Johnston (2008)) and MADELON ($2000 \times 500$) (Guyon (2004)) for the feature selection task using our proposed algorithm.

Analysis: - The numerical results presented in Table 10 clearly demonstrate that the SSVQR PI feature selection (as detailed in Algorithm 3) can significantly reduce the number of features while maintaining the quality of the PI, as measured by PICP and MPIW. On average, it could reduce the 69% of features on considered datasets and reduce the complexity of the PI estimation task significantly. The training time of the PI estimating task improves significantly after dropping irrelevant features through Algorithm 3. On average, it could improve the 53% improvement in training time reported in seconds. Also, reducing the significant numbers of features will reduce the tunning and testing time of the PI model.

For high-dimensional datasets such as UCI secom and MADELON, feature selection leads to a significant improvement in the quality of the estimated PI in terms of its coverage. The SSVQR based feature selection algorithm eliminates a significant number of irrelevant features and learn the PI in relatively much lower dimension which results in better PI coverage with linear functions. Table 11 lists the features dropped by the SSVQR-PI feature selection algorithm for each dataset.

| Dataset | Dimension | Before Feature Selection | | | After Feature Selection | | | % Reduced |
|---|---|---|---|---|---|---|---|---|
| | | PICP | MPIW | Train Time(sec.) | PICP | MPIW | Train Time(sec.) | Features |
| Spambase | (4601, 58) | 0.9663 | 0.9330 | 60 | 0.9653 | 0.9340 | 52 | 46% |
| Student Perf. | (395, 16) | 0.8861 | 6.8429 | 0.48 | 0.8861 | 6.8429 | 0.42 | 73% |
| Boston Housing | (505, 14) | 0.9406 | 23.0647 | 0.81 | 0.9406 | 23.0647 | 0.7 | 38% |
| uci-secom | (1568, 591) | 0.8758 | 1.6146 | 16 | 0.9204 | 1.6683 | 7 | 91% |
| MADELON | (2000, 500) | 0.7100 | 2.0007 | 58 | 0.9375 | 2.0020 | 11 | 99% |

Table 10: Performance comparison before and after feature selection using the SSVQR PI model. SSVQR PI feature selection method ( detailed in Algorithm 3) can significantly reduce the number of features while maintaining the quality of the PI, as measured by PICP and MPIW. Eliminating irrelevant features using Algorithm 3 significantly reduces the complexity of the PI estimation task. It also causes an average improvement of about 53% in Train Time (sec.) (Training time measured in seconds).

| Dataset | Dropped Features |
|---|---|
| Spambase | 2, 8, 11, 13, 16, 17, 18, 19, 20, 21, 22, 23, 25, 27, 31, 35, 37, 46, 47, 48, 49, 50, 53, 54, 55, 56 |
| Student Perf. | 1, 2, 3, 4, 5, 6, 7, 8, 9, 10, 11 |
| Boston Housing | 2, 3, 4, 7, 10 |
| UCI Secom | 1, 4-20, 22-27, 28-41, 42-50, 52-58, 60-66, 68-70, 72, 74-87, 89, 91-110, 112-132, 134, 137-138, 140-157, 159-160, 162-187, 189-203, 205-224, 226-246, 247-296, 297-332, 333-362, 364-386, 387-400, 401-418, 420-422, 424-431, 434-435, 437-438, 440-466, 467, 469-481, 483, 489-498, 501-509, 512-520, 522-538, 540-545, 546, 548-560, 563-569, 571, 573-579, 580, 582-587 |
| MADELON | 0-89, 91-227, 229-275, 277-331, 333-444, 446-499 |

Table 11: Features dropped by the SSVQR PI feature selection method ( detailed in Algorithm 3). On average, it drops the 69% of features on considered datasets and reduce the complexity of the PI estimation task significantly.

## 5.3 Probabilistic Forecasting with SVM models

In this section, we compare the performance of the SSVQR, SVQR and LS-SVR model for the probabilistic forecasting along with deep learning models.

Datasets:- We consider three popular time-series datasets namely Female Births ($365 \times 1$) (datamarket.com), Minimum Temperature ($3651 \times 1$) (machinelearningmastery.com) and Beer Production ($464 \times 1$) (Australian (1996)). We have used the 70% of dataset as training set and rest of them are used for testing. Out of the training set, the last 10% of the observations have been used for the validation set.

Baseline Methods:- We also train several recent and widely adopted Quantile based deep learning architectures for probabilistic forecasting developed in distribution-free setting, including Quantile-based LSTM, Quantile-based GRU, Quantile-based TCN and Quantile-based Transformers. The SVQR and SSVQR models are fundamentally quantile-based probabilistic forecasting approaches; therefore, when comparing them with deep learning counterparts, we should only consider quantile-based deep learning methods. However, for the sake of extensive comparisons, we also report the performance of deep probabilistic forecasting models that adopt different probabilistic forecasting approaches, including the Tube Loss LSTM and Quality-Driven (QD) Loss LSTM models (Pearce et al. (2018)), as well as the DeepAR model (Salinas et al. (2020)). The QD loss (Pearce et al. (2018)) is the improved version of the LUBE model which can be used minimized with the gradient descent method in deep learning architecture.

Experimental Setup:- The SSVQR, SVQR and LS-SVR model based probabilistic forecasting model were trained in MATLAB. All of the models were asked to obtain the probabilistic forecast for target calibration $1 - \alpha = 0.95$. For this, they estimate the $0.975^{th}$ and $0.025^{th}$ quantiles of the predictive distribution. All of them requires the tuning of the two parameters namely $C$ and RBF kernel parameter $\gamma$. We have tunned the value of the these parameters using the grid search in the search space $\{2^{-15}, 2^{-14}, ...., 2^{14}, 2^{15}\} \times \{2^{-15}, 2^{-14}, ..., 2^{14}, 2^{15}\}$.

The deep learning based probabilistic forecasting models such as Quantile-based LSTM, Quantile-based GRU, Quantile-based TCN, Quantile-based Transformers, Tube Loss LSTM, Deep AR and QD Loss LSTM

| Dataset | Method | PICP (mean) | PICP (std) | MPIW (mean) | MPIW (std) | Time(sec.) | Sparsity |
|---|---|---|---|---|---|---|---|
| FB. | SSVQR | 0.93 | 0 | 28.00 | 0 | 1.12 | 61 % |
| | **SVQR** | **0.95** | 0 | **27.11** | 0 | **0.97** | 0 % |
| | LS-SVR | 0.95 | 0 | 37.70 | 0 | 3.60 | 0 % |
| | Quantile LSTM | 0.95 | 0.02 | 28.20 | 2.15 | 118.00 | 0 % |
| | Quantile GRU | 0.95 | 0.01 | 46.52 | 3.13 | 4.00 | 0 % |
| | Quantile TCN | 0.96 | 0.01 | 57.00 | 6.16 | 17.29 | 0 % |
| | Quantile Transformer | 0.95 | 0.02 | 59.00 | 2.77 | 5.75 | 0 % |
| | Tube LSTM | 0.96 | 0.01 | 28.09 | 1.85 | 43.00 | 0 % |
| | QD LSTM | 0.94 | 0.03 | 38.98 | 3.25 | | 0 % |
| | DeepAR | 0.94 | 0.02 | 29.8 | 2.45 | 55.0 | 0 % |
| Mn Temp. | SSVQR | 0.96 | 0 | 10.72 | 0 | 200.79 | 69 % |
| | SVQR | 0.96 | 0 | 75.81 | 0 | 172.00 | 0 % |
| | **LS-SVR** | 0.95 | 0 | **10.59** | 0 % | **0.53** | 0 % |
| | Quantile LSTM | 0.95 | 0.02 | 24.82 | 1.95 | 1135.00 | 0 |
| | Quantile GRU | 0.94 | 0.02 | 16.53 | 0.39 | 37.63 | 0 % |
| | Quantile TCN | 0.95 | 0.01 | 16.84 | 0.26 | 184.07 | 0 % |
| | Quantile Transformer | 0.94 | 0.02 | 17.13 | 0.32 | 54.81 | 0 % |
| | Tube LSTM | 0.94 | 0.02 | 15.56 | 1.25 | 447.00 | 0 % |
| | QD LSTM | 0.79 | 0.04 | 5.94 | 0.45 | | 0 % |
| | DeepAR | 0.90 | 0.03 | 12.79 | 1.05 | 58.0 | 0 % |
| BP. | SSVQR | 0.96 | 0 | 78.21 | 0 | 1.77 | 70 % |
| | **SVQR** | **0.95** | 0 | **75.73** | 0 | **0.78** | 0 % |
| | LS-SVR | 0.96 | 0 | 79.48 | 0 | 0.29 | 0 % |
| | Quantile LSTM | 0.94 | 0.03 | 134.80 | 10.25 | 132.80 | 0 % |
| | Quantile GRU | 0.94 | 0.02 | 139.88 | 7.35 | 4.77 | 0 % |
| | Quantile TCN | 0.95 | 0.01 | 110.32 | 4.23 | 24.93 | 0 % |
| | Quantile Transformer | 0.94 | 0.02 | 124.90 | 4.41 | 7.72 | 0 % |
| | Tube LSTM | 0.95 | 0.02 | 42.91 | 3.45 | 89.60 | 0 % |
| | QD LSTM | 0.96 | 0.02 | 159.71 | 12.15 | | 0 % |
| | DeepAR | 0.76 | 0.04 | 12.43 | 0.95 | 62.0 | 0 % |

Table 12: Performance comparison of SVM and deep learning based probabilistic forecasting methods on benchmark datasets. Across five training runs, the SVM-based probabilistic forecasting models show zero standard deviation (std), indicating highly stable solutions, whereas deep learning models exhibit significant model uncertainty. SVM-based models still achieve competitive data uncertainty estimates as reflected in their PICP and MPIW values. FB: Female Birth, Mn.Temp.:- Minimum Temperature, BP.:- Beer Production and Time(sec.): Training time in seconds.

| Dataset | Model | Architecture | Number of weights | Window Size |
|---|---|---|---|---|
| Female Births | SVQR | Kernel | 256 | 10 |
| | SSVQR | Kernel | 256 | 15 |
| | LS-SVR | Kernel | 256 | 20 |
| | Quantile LSTM | LSTM [100] | 30 K | 25 |
| | Tube LSTM | LSTM [100] | 30 K | 25 |
| | QD LSTM | LSTM [100] | 30 K | 25 |
| | DeepAR | LSTM [40,40] | 13 K | 28 |
| | Quantile GRU | GRU [128] | 65 K | 12 |
| | Quantile TCN | TCN [32,64,64] | 120 K | 12 |
| | Quantile Transformer | Transformer [64,4,2] | 80 K | 12 |
| Minimum Temp. | SVQR | Kernel | 2 556 | 12 |
| | SSVQR | Kernel | 2 556 | 18 |
| | LS-SVR | Kernel | 2 556 | 22 |
| | Quantile LSTM | LSTM [16,8] | 32 K | 28 |
| | Tube LSTM | LSTM [16,8] | 32 K | 28 |
| | QD LSTM | LSTM [16,8] | 32 K | 28 |
| | DeepAR | LSTM [40,40] | 13 K | 56 |
| | Quantile GRU | GRU [128] | 65 K | 12 |
| | Quantile TCN | TCN [32,64,64] | 120 K | 12 |
| | Quantile Transformer | Transformer [64,4,2] | 80 K | 12 |
| Beer Prod. | SVQR | Kernel | 325 | 8 |
| | SSVQR | Kernel | 325 | 14 |
| | LS-SVR | Kernel | 325 | 18 |
| | Quantile LSTM | LSTM [64,32] | 29 K | 24 |
| | Tube LSTM | LSTM [64,32] | 29 K | 24 |
| | QD LSTM | LSTM [64,32] | 29 K | 24 |
| | DeepAR | LSTM [40,40] | 13 K | 48 |
| | QuantileGRU | GRU [128] | 65 K | 12 |
| | Quantile TCN | TCN [32,64,64] | 120 K | 12 |
| | Quantile Transformer | Transformer [64,4,2] | 80 K | 12 |

Table 13: Comparison of the complexity of used SVM and deep learning based probabilistic forecasting models on benchmark datasets. LSTM [16,8] means that the used LSTM model has two hidden layer containing 16 and 8 neurons respectively. While the SVM-based model delivers competitive, stable performance with zero model uncertainty only with a few hundred parameters, deep learning models require optimizing thousands of parameters.

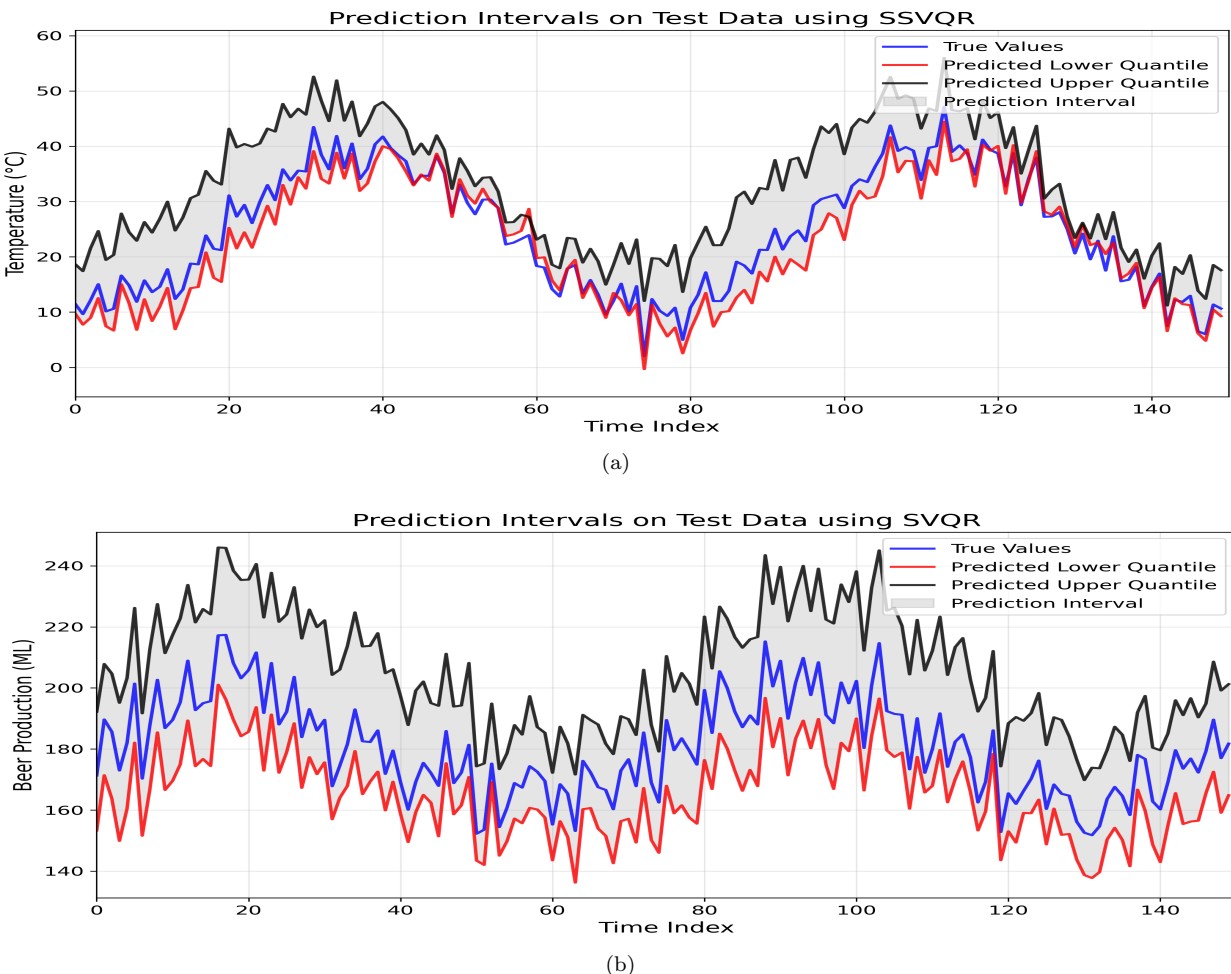

Figure 7: Probabilistic forecasting with SSVQR model on daily (a) Temperature and (b) Beer Production dataset

models were trained in a Python environment using the Tensor-Flow library. All of them were evaluated on the test set by effectively tuning their parameters on the validation set.

Analysis :- We present the numerical results in Table 12. All probabilistic forecasting models listed in Table 12 were trained five times using the same hyperparameter settings. It can be observed that the variance in the numerical results produced by the SVM-based probabilistic forecasting models is zero, indicating that these models exhibit negligible model uncertainty, whereas the deep learning models show significant model uncertainty. Furthermore, after proper parameter tuning, SVM-based probabilistic forecasting models achieve competitive PICP and MPIW values for quantifying data uncertainty, comparable to those produced by more complex LSTM-based deep forecasting models. Also, the SSVQR based probabilistic forecasting model obtains the sparse solution.

The tuned parameters of all probabilistic forecasting models used in the Table 12 are listed in Table 13.While the SVM-based model achieves competitive and stable performance along with zero model uncertainty with only a few hundred parameters to optimize, deep learning models typically require the optimization of thousands of parameters. Figure (7) (a) and (b) shows the forecasting of the SSVQR and SVQR model on Temperature and Beer Production datasets respectively.

### 5.4 Numerical Results of SVM based Conformal Regression method

We need to empirically show that the SVM based CR model estimates are more stable and less uncertain than NN based CR estimate.

| Dataset | Method | Performance | | | Std. (10 runs) | | |
| --- | --- | --- | --- | --- | --- | --- | --- |
| | | PICP (%) | MPIW | Time (sec.) | PICP | MPIW | Time (sec.) |
| Boston | CQR-NN | 94.12 | **2.19** | 2.65 | (1.14) | (0.12) | (0.23) |
| | SVQR+CP | **95.10** | 2.24 | **0.44** | (0.00) | (0.00) | (0.03) |
| Energy | CQR-NN | 87.66 | 1.13 | 3.20 | (1.12) | (0.03) | (0.31) |
| | SVQR+CP | **88.96** | **1.05** | **0.95** | (0.00) | (0.00) | (0.05) |
| Concrete | CQR-NN | 92.58 | 19.62 | 1.89 | (1.02) | (0.71) | (0.17) |
| | SVQR+CP | **91.35** | **18.74** | **0.38** | (0.00) | (0.00) | (0.01) |
| Yatch | CQR-NN | **91.69** | **2.43** | 1.44 | (1.49) | (0.35) | (0.12) |
| | SVQR+CP | 90.82 | 2.87 | **0.22** | (0.00) | (0.00) | (0.01) |
| Servo | CQR-NN | 88.39 | 0.73 | 1.15 | (2.21) | (0.10) | (0.09) |
| | SVQR+CP | **89.74** | **0.68** | **0.16** | (0.00) | (0.00) | (0.01) |

Table 14: Comprehensive comparison of CQR-NN and SVQR+CP methods across benchmark datasets. The SVM-based CR model (SVQR+CP) produces stable prediction sets with zero standard deviation, whereas the NN-based CR model (CQR-NN) exhibits substantial variability across training runs. Time (sec.):- Training time in seconds.

Baseline methods:- We compare performance of the SVM-based CR model (SVQR+CR) against the NN-based CQR model (CQR-NN) (Romano et al. (2019)) on benchmark datasets namely Boston Housing (Harrison Jr & Rubinfeld (1978)), Concrete (Yeh (1998)), Energy Efficiency (Tsanas & Xifara (2012)), Yatch Hydrodynamics (Gerritsma & Versluis (1981)) and Servo (Ulrich (1986)) under the split conformal setting, with the target coverage level $1 - \alpha = 0.90$.

Experimental Setup:- Both of the models SVQR+CR and CQR-NN were trained in Python. Both models were trained across 10 independent runs using fixed hyperparameter settings and identical splits of training and calibration sets. For the experiments, each dataset was partitioned into a training set (60%), a calibration set (20%), and a test set (20%). For the SVQR+CR model, we used the RBF kernel with parameter $\gamma = 1.0/n_{\text{features}}$, a regularization parameter $C = 10.0$, for targeting the $0.05^{th}$ and $0.95^{th}$ quantile function. For the CQR-NN model, the architecture consists of a dense layer with 200 neurons and ReLU activation, followed by a linear output layer. The model was trained for 200 epochs using the Adam optimizer with a learning rate of 0.01 and a batch size of 40, minimizing the pinball loss.

Analysis:- The numerical results in Table 14 yield several insights. The standard deviations of PICP and MPIW across the 10 runs were zero for SVM + CP, indicating perfectly stable predictions. In contrast, CQR-NN exhibited noticeable variability. This is expected, as neural network models, owing to their non-convex optimization landscape, may converge to different local minima across training runs, even with identical data and hyper-parameters.

Furthermore, SVM + CP achieved lower MPIW in 3 out of 5 datasets along with zero- parameter uncertainty. The CQR-NN method obtains the lower MPIW values in 2 out of 5 datasets however with significant parameter uncertainty. For quantifying the overall uncertainty, we should account both data uncertainty (MPIW values) and model uncertainty (parameter uncertainty). Clearly, the (SVM + CP) is less uncertain (more certain) than the CQR-NN.

# 6 Conclusions and Future Work

This paper presents a comprehensive roadmap for exploring UQ methods within the SVM regression framework, emphasizing its practical advantages in real-world applications. Due to the inherent stability and absence of parameter uncertainty in SVMs, the resulting PIs exhibit significantly lower overall uncertainty compared to those generated by NN. This makes SVMs a more reliable choice when accurate and stable PIs are critical for decision-making. Moreover, in probabilistic forecasting tasks, SVM based PI models can achieve comparable or even superior forecast quality while maintaining simpler model structures and requiring less training time. For high-dimensional data, we show that incorporating feature selection before training substantially improves both the PI quality and computational efficiency. In the context of conformal prediction for regression, our findings indicate that SVM models produce more stable and less uncertain prediction sets than NNs, making them a better alternative in such settings.

We develop a feature selection algorithm for PI estimation under the assumption that the bounds are linear functions of the input features. However, extending this approach to handle non-linear dependencies remains an important direction for future work, particularly in both NN and SVM-based models. Additionally, we show that SVM-based probabilistic forecasting models offer a compelling alternative to complex deep learning architectures by significantly reducing model complexity through the optimization of fewer parameters on batch datasets. Despite these advantages, they are not well-suited for dynamic or online data scenarios. To address this limitation, developing incremental or online SVM-based probabilistic forecasting models presents a promising avenue for future research.

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
