# OpenReview forum: "Uncertainty Quantification in SVM prediction"
_TMLR — Rejected by TMLR_

### Review · Reviewer_5Nrn · 2025-09-24

**Summary Of Contributions:**

This paper brings sparsity and conformal guarantees into SVM-based uncertainty quantification, backed by both theoretical grounding and experimental validation. Its main limitations lie in scalability, dataset diversity, and depth of comparisons with deep learning models. The proposed method, Sparse Support Vector Quantile Regression (SSVQR), has not been tested with very large datasets (common in modern applications). The paper claims that SVMs perform comparably/superior to deep nets are promising, but the evaluation is limited to a few architectures and does not reflect the latest state-of-the-art models. While the paper ensures convexity and sparsity, it provides less formal analysis of convergence rates or generalization bounds for SSVQR.

**Audience:**

Yes

**Audience Explanation:**

There may be some readers who will be interested in this work, but not many.

**Claims And Evidence:**

No

**Claims Explanation:**

As mentioned earlier, the paper claims that SVMs perform comparably/superior to deep nets are promising, but the evaluation is limited to a few architectures and does not reflect the latest state-of-the-art models. While the paper ensures convexity and sparsity, it provides less formal analysis of convergence rates or generalization bounds for SSVQR.

**Requested Changes:**

The evaluation could be strengthened by incorporating a broader range of representative state-of-the-art neural network architectures, ensuring a more comprehensive and fair comparison against contemporary deep learning methods. In addition, a more formal theoretical analysis of the SSVQR framework—particularly regarding convergence rates and generalization bounds—is recommended to enhance the rigor and provide stronger guarantees of its performance across diverse settings.

---

> ### Author Response · Authors · 2025-09-27
> **More comprehensive and fair comparison against contemporary deep learning methods  (part 1 )**
>
> Thanks for your comment.  We agree that our one of the claim is that the SVM based probabilistic forecasting models also possess the potential of obtaining the good quality probabilistic forecast. In our experiments, we have shown that the SVM based probabilistic forecasting model can obtain the superior or comparable results over the recent deep probabilistic forecasting models.
>
>
>  Please note that we don’t claim that the SVM has better ability to extract the meaningful temporal features than deep learning model architectures. The deep learning model are complex architectures and hence when the autoregressive relations are very complex in particular way along with availability of the large scale data, the deep learning models are expected to perform better.  However, it is not always the case. The literature has shown that even a simple AR or ARIMA model can obtain better point forecasting over the complex deep learning architecture in several cases.
>
> Returning back to our UQ context, our claim regarding the comparable performance of the SVM model over the deep leaning model is not upon the basis of the ability of capturing the complex auto-regressive relationship, though it is because of the nature of their solution they produce.
>
> The deep learning architecture converges to the different solution and hence the forecasting results in different trail and the variance of their solutions increases as the sample size becomes moderate. It causes the model uncertainty to grow significantly for the deep learning model. The overall uncertainty requires to be in effective decision making should include the model uncertainty and data uncertainty.    However, please note that there is no any concrete and well established method for capturing the model uncertainty of the deep learning model.  Following the (Laxinanaryan et al., 2017) , (Pearce et al.,2018) and (Salem et al. (2020),  one can use the ensembles and obtain the overall uncertainty by adding the parameter uncertainty with the data uncertainty.  However, their formulation for measuring the parameter uncertainty arbitrary divides variance of the parameters by m (number of training trails) without any justification while, they agree for actually measuring the model uncertainty, it should not be divided by m (see the discussion on the page 4 (section 2.4.2)of Salem et al. and  also see  , (Pearce et al.,2018))
>
> On the other hand, the SVM exhibits the zero parameter uncertainty due to its global optimal solution. Its makes the overall uncertainty reduces to the data uncertainty in case of the SVM. Please note that, in Table 12 of the manuscript, we have reported only the data uncertainty for each of the evaluated models, as in case of the deep learning model, the literature lacks the clarity how to fuse the model uncertainty and data uncertainty in the final PI evaluation.  If we use the Ensemble approach used by the (Pearce et al.,2018) and (Salem et al. (2020))., without dividing the variance of parameters optimized in different random trail by total number of trail m, the model uncertainty will increase significantly and it will reflect different picture.  However, we have kept these things implicit in our research writing and only claim that the parameter uncertainty in case of the SVM remains zero.
>
> In Table 12, we have compared the SVM models with very recent deep forecasting models used for the probabilistic forecasting on three different benchmark datasets. We have used the Tube loss based deep forecasting model (Tube LSTM) (developed in 2024) , Quantile based deep forecasting model (Quantile LSTM) (popularly used recently in energy domain) , Deep AR model (popular model developed in 2017)  and QD loss LSTM model  (developed around 2020) as baseline methods.  All of our baseline models are recently developed. We refrain ourselves to compare the old distribution based probabilistic forecasting models as, we are performing the comparison with the distribution free SVM based Pinball loss models.
>
>
> References:-
>
> Balaji Lakshminarayanan, Alexander Pritzel, and Charles Blundell. Simple and scalable predictive uncertainty estimation using deep ensembles. Advances in neural information processing systems, 30, 2017.
>
>  Tim Pearce, Alexandra Brintrup, Mohamed Zaki, and Andy Neely. High-quality prediction intervals for deep learning: A distribution-free, ensembled approach. In International conference on machine learning, pp. 4075–4084. PMLR, 2018.
>
> Salem, T. S., Langseth, H., & Ramampiaro, H. (2020, August). Prediction intervals: Split normal mixture from quality-driven deep ensembles. In Conference on Uncertainty in Artificial Intelligence (pp. 1179-1187). PMLR.

---

> ### Author Response · Authors · 2025-09-27
> **More comprehensive and fair comparison against contemporary deep learning methods (part 2 )**
>
> We can compare the complex deep forecasting architectures such as transformer on these datasets but, it cannot help us to reduce the data uncertainty ( in terms of MPIW) values significantly. Also, the complex models involve more number of parameters which again increases the parameter uncertainty particularly on moderate and small scale datasets.
>
>  Also, we agree that on large-scale datasets, deep learning models may show the lesser parameter uncertainty and also capture the temporal features well for better probabilistic forecast.  SVM models may not show the similar degree of the scalability for very large scale datasets.  But, please note that the objective of our research is not to replace the deep learning models with the SVM model for UQ task. We only want to emphasize that the SVM models are less uncertain, more trustworthy and obtain the sparse solution than NN. Therefore, the UQ task, researchers should also consider the SVM model as an alternative choice of modelling in several conditions particularly for small and moderate scale dataset.
>
> We think that we should add a brief summary of the above discussion in the revised manuscript to better clarify the story to the readers. Further, we shall be happy to add the numerical benchmark in  Table  12 as per  your suggestions.
>
>
> This study of the SVM model from the lens of uncertainty has the following straightforward technical impacts and practical use for readers.
>  1. In real-world decision making, if you want to obtain the PI, you may prefer the SVM model. It is because that as compared to NN, the overall uncertainty (including the model uncertainty) of the SVM-based PI model is always lesser due to zero parameter uncertainty.
>
> 2. Also for probabilistic forecast, the SVM-based PI model may help you to obtain the similar or better quality probabilistic forecast with less training time and simpler structure, particularly for small and moderate size datasets.
>
> 3. If you are working with the high dimensional data, consider the feature selection task before training for the PI to improve the PI quality and complexity of the model.
>
> 4. If you want to obtain the conformal prediction set for regression task , you should work with SVM model as opposed to NN. The SVM-based conformal prediction set is more stable and less uncertain.
>
> 5. Apart from the above practical use, this research also provides the road-map for the development of the UQ method in SVM framework

---

> ### Author Response · Authors · 2025-09-27
> **Formal theoretical analysis of the SSVQR framework—particularly regarding convergence rates and generalization bounds**
>
> The SSVQR model minimizes the pinball loss function with an added regularization term. The asymptotic convergence properties of quantile regression( pinball loss)  with kernel-induced functions under various choices of regularization are rigorously studied in (Takeuchi et al., 2006). In addition, (Takeuchi et al., 2005) establishes a meaningful theoretical risk bound under general conditions (Theorem 6) for the pinball loss minimizer. Further, it derives a bound on the expected quantile (Theorem 7). We humbly submit that we could not find  any space to generate any theorem and its proof.
>
> Regarding the convergence rate: -
>
>  The SVQR problem (2) and SSVQR problem (16) of the manuscript, both are convex function of the w and b,  as $w^T\phi(x)+b$ is a function generated by positive-semi definite kernel . The SVQR problem (2) is a Quadratic Programming Problem (QPP) and problem (16) is a Linear Programming Problem (LPP) and they can be solved with any optimization algorithm available in the literature for their global solution. In our experiments, we have used the ‘quadprog’ and ‘linprog’ packages to solve the QPP and LPP which uses the ‘interior-point-convex ’ and ‘simplex method ‘ algorithms respectively. We humbly submit that we have not developed any new optimization algorithm for solving the quantile problems hence we have not found any space to add any theorem regarding this.
>
> References: -
> Takeuchi, I., Le, Q., Sears, T., & Smola, A. (2006). Nonparametric quantile estimation, MIT press

---

> ### Author Response · Authors · 2025-10-18
> **Including additional neural architectures for benchmark comparison**
>
> As per your suggestion, we have included the performance of the Quantile GRU, Quantile TCN, and Quantile Transformer models as additional benchmark methods in Table 12 of our manuscript. The SVQR and SSVQR models are fundamentally quantile-based probabilistic forecasting approaches; therefore, when comparing them with deep learning counterparts, we should only consider quantile-based deep learning methods. However for the sake of comparative study, we have included the performance of the recent deep probabilistic forecasting models that utilizes different  probabilistic forecasting approaches.
>
>
> | Dataset | Method | PICP | PICP (std) | MPIW | MPIW (std) | Training Time | Sparsity |
> |:---------|:--------|:------:|:-----------:|:------:|:-------------:|:--------------:|:-----------:|
> | **Female Births** | SSVQR | 0.93 | 0 | 28.00 | 0 | 1.12 | 61% |
> |  | **SVQR** | **0.95** | 0 | **27.11** | 0 | **0.97** | 0% |
> |  | LS-SVR | 0.95 | 0 | 37.70 | 0 | 3.60 | 0% |
> |  | Quantile LSTM | 0.95 | 0.02 | 28.20 | 2.15 | 118.00 | 0% |
> |  | ⭐ Quantile GRU | 0.95 | 0.01 | 46.52 | 3.13 | 4.00 | 0% |
> |  | ⭐ Quantile TCN | 0.96 | 0.01 | 57.00 | 6.16 | 17.29 | 0% |
> |  | ⭐ Quantile Transformer | 0.95 | 0.02 | 59.00 | 2.77 | 5.75 | 0% |
> |  | Tube LSTM | 0.96 | 0.01 | 28.09 | 1.85 | 43.00 | 0% |
> |  | QD LSTM | 0.94 | 0.03 | 38.98 | 3.25 | — | 0% |
> |  | DeepAR | 0.94 | 0.02 | 29.8 | 2.45 | 55.0 | 0% |
> | **Minimum Temp.** | SSVQR | 0.96 | 0 | 10.72 | 0 | 200.79 | 69% |
> |  | SVQR | 0.96 | 0 | 75.81 | 0 | 172.00 | 0% |
> |  | **LS-SVR** | **0.95** | 0 | **10.59** | 0 | **0.53** | 0% |
> |  | Quantile LSTM | 0.95 | 0.02 | 24.82 | 1.95 | 1135.00 | 0% |
> |  | ⭐ Quantile GRU | 0.94 | 0.02 | 16.53 | 0.39 | 37.63 | 0% |
> |  | ⭐ Quantile TCN | 0.95 | 0.01 | 16.84 | 0.26 | 184.07 | 0% |
> |  | ⭐ Quantile Transformer | 0.94 | 0.02 | 17.13 | 0.32 | 54.81 | 0% |
> |  | Tube LSTM | 0.94 | 0.02 | 15.56 | 1.25 | 447.00 | 0% |
> |  | QD LSTM | 0.79 | 0.04 | 5.94 | 0.45 | — | 0% |
> |  | DeepAR | 0.90 | 0.03 | 12.79 | 1.05 | 58.0 | 0% |
> | **Beer Prod.** | SSVQR | 0.96 | 0 | 78.21 | 0 | 1.77 | 70% |
> |  | **SVQR** | **0.95** | 0 | **75.73** | 0 | **0.78** | 0% |
> |  | LS-SVR | 0.96 | 0 | 79.48 | 0 | 0.29 | 0% |
> |  | Quantile LSTM | 0.94 | 0.03 | 134.80 | 10.25 | 132.80 | 0% |
> |  | ⭐ Quantile GRU | 0.94 | 0.02 | 139.88 | 7.35 | 4.77 | 0% |
> |  | ⭐ Quantile TCN | 0.95 | 0.01 | 110.32 | 4.23 | 24.93 | 0% |
> |  | ⭐ Quantile Transformer | 0.94 | 0.02 | 124.90 | 4.41 | 7.72 | 0% |
> |  | Tube LSTM | 0.95 | 0.02 | 42.91 | 3.45 | 89.60 | 0% |
> |  | QD LSTM | 0.96 | 0.02 | 159.71 | 12.15 | — | 0% |
> |  | DeepAR | 0.76 | 0.04 | 12.43 | 0.95 | 62.0 | 0% |
> **Table: Performance comparison of SVM and deep learning based probabilistic forecasting methods on benchmark datasets.**
>
> All probabilistic forecasting models were trained five times using the same hyperparameter settings. It can be observed that the variance in the numerical results produced by the SVM-based probabilistic forecasting models is zero, indicating that these models exhibit negligible model uncertainty, whereas the deep learning models show significant model uncertainty.  The tuned parameters of all probabilistic forecasting models used in the above table are listed in Table 13 of the revised manuscript. While the SVM model involves optimization of only a few hundred weights, deep learning models require optimization of thousands of weights.
>
> We humbly request you to provide your further suggestions or query if any. After listening from you, we shall submit the revised manuscript incorporating your suggested (above mentioned) changes before the 5th November 2025. Looking for your cooperation and response

---

### Review · Reviewer_Vwt6 · 2025-09-29

**Summary Of Contributions:**

This paper contributes an overview of the field of uncertainty quantification and a novel method for SVM-based models with solvable optima. This work concretely proposes that the addition of an L1-norm on parameters of an SVM for a Quartile regression problem (and with appropriate re-parameterization) can allow for exact ‘sparse support vector quantile regression’ (SSVQR) solutions for application to predictive interval estimation and probabilistic forecasting. This new model is then tested and compared against existing SVM/SVQR-based models as well as to uncertainty quantifying neural network models. Ultimately, it is proposed that the clarity of the optima of these SV-based methods might compete with and, at times, outperform alternative methods.

**Audience:**

Yes

**Audience Explanation:**

Yes, I do believe that a number of TMLR readers would be interested in the findings of this work, though in it’s current state I do not believe that the findings are readable (see comments below).

**Claims And Evidence:**

No

**Claims Explanation:**

**The evidence for some of the claims made in this work are unclear. I would go so far as to say that some are also unconvincing.** The SSVQR method described is, if anything, marginally different from the regular SVQR method (single additional regularization). In the six artificial datasets tested, the comparison is also very difficult to draw, with almost all measures showing little difference between SSVQR and SVQR. To be clear, this is not a bad thing, but it is unclear whether and in which conditions SSVQR is a necessary or beneficial addition for any particular application. A question I am left with is, why and when should I use the SSVQR method?
One can take away that SSVQR appears to produce more sparse models and often requires less time to train (at least, that is what I assume ‘Time (s)’ refers to) but why this training is faster is also unclear and not consistent across experiments.
Furthermore, in probabilistic forecasting tasks (Tables 12), the other SV methods (SVQR and LS-SVR) outperform SSVQR consistently, both in accuracy and in training time, and in two of three tasks the NN based methods even outperform all SV-based methods. All in all, the take-aways of this work are a little unclear. Indeed these analytically solvable models have a number of benefits but I do not believe that those were well demonstrated here.

**Requested Changes:**

The set of requested changes for this work are innumerable. Unfortunately, the quality of the writing is simply too poor for me to provide a point-by-point set of changes. From the very first page, I have more than a half-dozen notes on grammatical errors and missing or incorrect words/terms/tenses. I would recommend that the authors examine the quality of the written text in detail and perhaps even employ a proof-reader with good skill to work through the text. Giving point by point feedback on this is not feasible for me as reviewer.

Beyond the writing quality, this work suffers from a lack of clear communication of methods and results. The tables provided in this work contain a number of columns whose headings are never specified. For example, “Spar (Lw, Up)” in Tables 2-9 is never defined in the text. Other acronyms are defined well before the tables and must be recalled, or searched for, to parse the tables of results. The text around the tables and table captions are insufficient for a reader to really follow the results without having to interpret a great deal. Even things as significant as the datasets used have no references, for example the “Female Births”, “Minimum Temperature” and “Beer Production” datasets have no citations or context. Furthermore, key metrics like ‘Training Time’ are sometimes also referred to as ‘Time (s)’ and an explanation of precisely the meaning/measurement (or in some cases units) of these metrics are just missing. Finally, many of the results could also have been better presented as plots or placed in appendices to aid the flow and communication of the paper.

**All in all, for these reasons I cannot recommend an acceptance of this paper in its current state. A huge amount of rewriting would have to be carried out before I could seriously provide a request for changes to it's scientific content.** In general, the concerns I have around the claims (above) would have to be addressed also to give this work clarity in communication of it's takeaways.

---

> ### Author Response · Authors · 2025-10-18
> **Requested Changes: Improving the readability of the paper**
>
> We are sorry that you found the readability of the paper to be poor due to grammatical, spelling, and sentence tense structure errors.
> We have thoroughly reviewed the entire content of the paper and conducted proofreading using both human and AI assistance. While there were a few grammatical issues, such as subject-verb disagreements, pluralization errors, and minor spelling mistakes, these have all been corrected. The revised manuscript is now free of any grammar or spelling errors.
>
> In the revised manuscript, we have clearly defined all acronyms used in the table headings within the respective captions. Furthermore, we have enriched each table caption with sufficient descriptions to help readers understand the significance of the numerical results without needing to refer back to the main text. References for all datasets used in our experiments have also been included. We had used the acronym 'Time (s)' to denote the training time of the model in seconds, but, this has been explicitly clarified in the corresponding table captions to enhance readability in the revised manuscript. We have incorporated all of your valuable suggestions, revised the manuscript accordingly, and enhanced its readability.
>
> We humbly submit that this study of the SVM model, viewed through the lens of uncertainty quantification, is valuable and has the following straightforward technical impacts and practical uses for readers.
>
> 1.	In real-world decision-making, if one wishes to obtain the PI, the SVM model may be preferred. This is because, compared to NN, the overall uncertainty (including model uncertainty) of the SVM-based PI model is lower due to zero parameter uncertainty.
>
> 2.	Also, for probabilistic forecasting, the SVM-based PI model may help obtain a similar or better-quality probabilistic forecast with less training time and a simpler structure, particularly for small and moderate-sized datasets.
>
> 3.	If you are working with high-dimensional data, consider performing feature selection before training for the PI to improve its quality and reduce the model’s complexity.
>
> 4.	If you want to obtain a conformal prediction set for a regression task, you should work with the SVM model rather than the NN. The SVM-based conformal prediction set is more stable and exhibits lower uncertainty.
>
> 5.	Apart from the above practical uses, this research also provides a roadmap for the development of UQ methods within the SVM framework.
>
> We humbly request you for your further response and queries if any.  After listening from you, we shall submit the revised manuscript  incorporating the above changes before the 5th November 2025. Looking for your co-operation and response.

---

> > ### Comment · Reviewer_Vwt6 · 2025-11-04
> >
> > I'm still awaiting an updated copy of this manuscript so that I can review it again. I hope also that my points around clarity and the supporting of claims with clear evidence have been taken into account.

---

> > > ### Author Response · Authors · 2025-11-05
> > >
> > > Thanks for your reply. I shall update the copy of the revised manuscript in the light of the changes, you have indicated. It may take some time. I shall try to update by 15 November.

---

> > > > ### Author Response · Authors · 2025-11-16
> > > >
> > > > Thank you for your valuable comments. They have significantly improved the overall quality of the
> > > > manuscript.   As per your suggestions, we have revised and updated the manuscript.
> > > >
> > > > We have also included, as supplementary material, a detailed response-to-reviewers document that provides point-by-point answers to each of your comments and queries, along with the revised manuscript where all changes are highlighted in blue. Additionally, we have explicitly mapped each of our claims to the corresponding empirical evidence presented in the paper. We kindly request you to review these materials for clarity and completeness.

---

> ### Author Response · Authors · 2025-10-18
> **Regarding the Sparsity and Training Time**
>
> A sparse vector is one in which only a few components are non-zero. Consider the training set ${(x_i,y_i):x_i \in \mathbf{R}^n, y_i \in \mathbf{R}, i=1,2,\dots,m}$. For a linear regression model $w^Tx+b$, if $w \in \mathbf{R}^n$ is sparse with indices $I_1 \subseteq \{1,\dots,n\}$ equal to zero, then only the features in $\{1,\dots,n\} - I_1$ are relevant, and the remaining features can be eliminated. This elimination of the features does not increase the generalization ability and interpretability of the model but also reduces the overall computational time including the training, testing and parameter tunning time.
>
>  Further, for non-linear estimation using a kernel-based function $\sum_{i=1}^{m} k(x_i, x)u_i + b$, sparsity arises in the coefficient vector $u \in \mathbf{R}^m$. Even if certain indices $I_2 \subseteq \{1,2,\dots,m\}$ are zero, this does not eliminate input features of $x \in \mathbf{R}^n$. Instead, it means that the corresponding training points in $I_2$ do not contribute to constructing the regression function. Consequently, the regression can be expressed as $\sum \limits _{j \in \{1,2,\dots,m\} - I_2} k(x_j, x)u_j + b$. In this case, sparsity reduces the overall model complexity, leading to lower computational cost during training, testing and parameter tunning time and  improved generalization on unseen data.
>
>
> Due to its sparse solution vectors, the SSVQR-PI model is capable of performing feature selection, leading to a simpler and more interpretable solution. Under identical hyperparameter settings, the SSVQR-PI model is theoretically expected to require slightly less training time than the SVQR model. However, in practice, this also depends on the efficiency of the optimization solvers quadprog’ and ‘linprog’ used for solving the QPPs and LPPs of the SVQR and SSVQR-PI models, respectively. Our experimental results reported in Tables 3–9 are consistent with this expectation.
>
> In the earlier version (Table 2), the training times for the SVQR and SSVQR models were unintentionally reported inaccurately, as background processes running on the system affected the measurements. This has been corrected in the revised manuscript. In Table 12, the training time of the SSVQR model appears higher than that of the SVQR-PI model because both were trained under different hyperparameter settings. Specifically, the window size tuned for the SSVQR model was larger than that of the SVQR-PI model (refer to Table 13 of the manuscript for the tuned parameters)
>
> For the probabilistic forecasting experiments, the model achieving a lower MPIW value (i.e., a narrower prediction interval) without compromising the PICP which should meet or exceed the target level of 0.95 is considered the best. We have included additional benchmark deep learning models as baselines in Table 12 of our manuscript (see our response to the last reviewer Reviewer 5Nrn (https://openreview.net/forum?id=LrazQEG1QW&noteId=wVorQ177tA) ). The SVM-based models obtains better performance compared to the quantile-based deep probabilistic forecasting models. While the SVQR and SSVQR models exhibit comparable performance, the latter offers a sparse and more interpretable solution. Another important advantage of using SVM-based models over deep learning models is that the latter exhibit significant model uncertainty, whereas SVMs demonstrate negligible or nearly zero model uncertainty.

---

### Review · Reviewer_cSjq · 2025-10-08

**Summary Of Contributions:**

The authors reviews and experimentally compares different ways to express uncertainty in regression problems.

The main contributions are

1. Formulation of a new method to produce uncertainty intervals for kernel SVM.
2. Showing that the underlying optimization problem can be reduced to a linear programming problem. This is an improvement since the earlier method of Takeuchi amounted to quadratic optimization problem.
3. Experimentally evaluating the method to show that is performs equally or better than previously proposed methods.

**Additional Comments:**

None.

**Audience:**

Yes

**Audience Explanation:**

Uncertainty quantification is an important question for regression problems.

**Broader Impact Concerns:**

None.

**Claims And Evidence:**

No

**Claims Explanation:**

In order to give credence to experimental claims to the paper, the author(s) should provide source code for their experiments.

The paper contains unsubstantiated claim about theoretical performance of the method, namely the following sentence:

"The asymptotical properties of the SSVQR model remain similar to the SVQR model detailed in (Takeuchi
et al. (2006))."

This is not sufficient. One should be extremely careful with probabilistic claims about SVM, since it is known that sparsity contradicts probability predictions in classification context  (Bartlett & Tewari, JMLR, 2007). Although the conflict has been described in classification context, this paper considers that the approach would be similar :"In this work, we have focused exclusively on the SVM regression
model, a similar UQ analysis can be extended to SVM classification models in future research.".

**Requested Changes:**

Critical for acceptance:

1. Provide an anonymized source code repository.
2. Provide clear guidance how to tune hyper parameters of the SSVQR method. This should be done in a separate section, because it is a critical but unspecified part of Algorithm 2.
3. Naming of section 2 should be changed. I propose "Preliminaries".

Strengthen the paper
- Not all readers will have access to (expensive) Matlab environment. If feasible, instructions how to modify the code for Octave should be included.
- The paper's main contribution is proposing a new method. The claim that it provides a roadmap is not accurate. It would be beneficial to shorten the paper with emphasis on new contributions and how they improve on state-of-the art.
- It is not clear why 6 synthetic datasets were evaluated, but the synthetic dataset from Takeuchi's paper was not among them. Generally, when a new method is evaluated, it should be compared also on the datasets discussed in the work proposing earlier method.
- Takeuchi's paper has discussion of monotonicity and non-crossing properties of quantile estimates. If appropriate for this method, these should be discussed as well.
- max operator should not italicized - please use $\max$ in LaTeX.

---

> ### Author Response · Authors · 2025-10-19
> **Requested Changes:**
>
> Critical for acceptance:
>
> 1.  As per your suggestion, we have provided the code and datasets used in our experiments at  https://github.com/anonymousmyaccount/myaccount/tree/main/-PI-IN-SVM--my_code . We shall incorporate it into our revised manuscript.
>
> 2. We humbly submit that we had provided the details regarding the parameter tuning in our submitted manuscript for each of the experiments separately.
>
> For artificial datasets, we describe
>  " The SVQR problem(4) or  SSVQR (18) problem requires the supply of the two user defined parameters $C$ and  RBF kernel parameter $\gamma$ for non-linear PI estimation. We have tunned the value of the these parameters using the grid search in the search space {$\{2 ^{-8},2^{-7},....,2^7,2^8 \}$}  $\times$  {$\{2 ^{-8},2^{-7},...,2^7,2^8 \} $}. "  in Section 5.1 under "Experimental Setup and Parameter Tunning" .
>
> In Section 5.2 for Feature selection experiments, we detail the following in submitted manuscript.
>
> "Following Algorithm 3, we perform feature selection using the SSVQR PI model for  $\bar{q} =0.025$ and present the results in Table 10. Since Algorithm 3 solves the problem 18 with linear kernel twice, we need to tune only the parameter $C$ for each $\bar{q}$  which was selected from the grid of $\{2^i: i = -15,....,15 \}$."
>
> In Section 5.3 for probabilistic forecasting experiments, we have given the following details in the submitted manuscript.
>
> " All of them (SVM models) requires the tuning of the two parameters namely $C$ and RBF kernel parameter $\gamma$. We have tunned the value of the these parameters using the grid search in the search space {$\{2 ^{-15},2^{-14},....,2^{14},2^{15} \}$} $\times$  {$2 ^{-15},2^{-14},...,2^{14},2^{15}  $.}"
>
> In Section 5.4 for conformal regression experiments, we have also provided the tunning parameter details.
>
>
> 3.  We have changed the Subsection 2 heading with the "Preliminaries" as suggested by you.
> --------------------------------------------------------------------------------------------------------------------------------------------------------------------------------
>
> Strengthen the paper:-
>
> *  In our code and data provided at https://github.com/anonymousmyaccount/myaccount/tree/main/-PI-IN-SVM--my_code , we have also included the python implementation of our and existing models. Also our MATLAB codes can be runed in the Octave environment as well .
>
> *   We have considered several artificial datasets in which we introduced different types of noise, including Normal, Uniform, and Chi-square distributions with varying parameters. The objective was to examine how the existing and proposed SVQR models perform under different noise distributions and variance scales. To this end, we generated six distinct artificial datasets for analysis.
> In contrast, Takeuchi et al. employed a relatively simple artificial dataset and evaluated the performance of the SVQR model only under a Normal noise distribution assumption. They have generated the artificial dataset by adding the normal noise in simple sinc function.  We can also include the evaluation of the existing and the developed SSVQR models on the same dataset if you wish; however, this would result in redundant experiments.
>
> * We respectfully submit that the discussion on the monotonicity and non-crossing properties of quantile estimates is not directly relevant to the scope and contributions of our work.
>
> * We have changed the style of the **max** operator in our manuscript.
>
>
> *  We humbly submit that  our paper main contribution is not limited to developing the SSVQR model rather, it is a study that aggregates the different relevant  concepts for studying the SVM models from the lens of the UQ in view of the recent developments carried out in the NN framework.
>
>
> We humbly request you for your further response and queries if any. After listening from you, we shall submit the revised manuscript incorporating your suggestions before the 5th November 2025. Looking for your co-operation and response.

---

### Comment · Action_Editor_ucTd · 2025-10-22

Dear Author,

You posted your responses on 19 October. Please give the referees time to review them before sending follow-up messages such as “Thank you for your reviews. I look forward to your consideration and reply.”

---

### Decision · Action_Editor_ucTd · 2025-11-24

**Recommendation:** Reject

**Audience:**

Yes

**Audience Explanation:**

Both UQ and SVMs are of interest to TMLR's audience.

**Claims And Evidence:**

No

**Claims Explanation:**

After careful consideration of the revised manuscript and the accompanying referee reports, I am afraid we are not able to accept your paper for publication in its current form. While the topic of sparse support vector quantile regression for uncertainty quantification is of potential interest and one referee notes that the method may be worth publishing once key issues are resolved, all the referees raise substantial and overlapping concerns about the current manuscript. They find the presentation and narrative difficult to follow, with the core methodological contribution described only late in the paper and its contribution over existing SVQR methods insufficiently justified. The empirical evidence does not convincingly support the claimed advantages: in several cases the proposed SSVQR is not clearly superior to established baselines, and key evaluation metrics (such as the width of prediction intervals) rely on assumptions about non-crossing quantiles that are not enforced in the implementation. In addition, the provided code and experimental setup are incomplete and not fully consistent with the results reported in the paper, creating serious doubts about reproducibility. Finally, important conceptual and theoretical points raised in earlier reviews (including the treatment of non-crossing quantiles, experiments on standard benchmark datasets, and the asymptotic properties of the method) have not been addressed in a sufficiently clear and well-substantiated way. For these reasons, I do not believe the manuscript can be brought to the level required for publication through a further round of revision, and I must decline publication.

---

> ### Author Response · Authors · 2025-12-01
>
> Dear Action Editor,
>
>
> Thanks for your continuous efforts in evaluating the paper and finalizing the decision. However, I have a different perspective about the overall summarization of the reviews which I think I need to express. I request you to please go through the followings.
>
>
> About the presentation: -
>
>
> The presentation of the paper is bit different because the nature of study it carries out.  The author as well as reader will suffocate a lot if you force it to arrange it in form of a typical paper (Introduction, Proposed Section, Theoretical Results and Experiments with recent benchmarks).
>
> The very basic idea of this paper has emerged when  I realize that the  $\textbf{SVM models show zero model uncertainty}$ and  it is a matter of surprised that why SVM based UQ methods were ignored in literature particularly in regression setting.  I think that this research gap in literature requires the urgent attention of UQ researcher.
>
> In above context, I have started the story with recently developed NN based UQ methods and highlights their limitations in term of their instable solution exhibiting the significant model uncertainty. We simply argue that the SVM based UQ models are less uncertain as they do not inherit any model uncertainty due to their global optimal (stable) solution. And this realization does not require framing any complex mathematical lemma and corresponding proof.
>
> Further, we systematically study the SVM based Quantile literature and emphasizes the benefits of obtaining the sparse solution. Thereafter, we modify the SVQR model a bit to obtain the sparse solution (SSVQR model). $ \textbf{Please note that the major core contribution of this paper is not only about developing the SSVQR model as it is the extension of the SVQR model.}$
>
> The major significance of this work lies in following points.
>
> (1)	How does the feature selection problem become important in PI estimation particularly for high dimension datasets?  How we can select the feature with the SVM based feature selection algorithm to improve the PI Quality and simplify the PI estimation task?
>
> (2)	How to obtain a stable Conformal prediction set in regression problem?  Modern NN based Conformal prediction set are very unstable however the SVM based Conformal set is stable, bit interpretable and simple due to their nature of solution.
>
> (3)	   How to obtain the stable prediction in probabilistic forecasting problem? For small and moderate scale datasets, we show that quality of the PI estimated by the SVM based models were comparable and sometime superior to the deep learning model. Also, the SVM based probabilistic forecasting models are simple, stable and show zero model uncertainty. It gives the SVM model an edge over the NN based probabilistic forecasting models particularly for the small and medium level datasets.
> I again repeat the major contribution of this work is not about developing the SSVQR model rather it lies the SVM applications in different UQ task and analyzing the corresponding advantages due to the nature of their solution.  I think this confusion has impacted the evaluation a lot even after my effort in overall review process.
>
> I again repeat the major contribution of this work is not about developing the SSVQR model rather it lies the $ \textbf{SVM applications in different UQ task and analyzing the corresponding advantages due to the nature of their solution.}$  I think this confusion has impacted the evaluation a lot even after my effort in overall review process.
>
> Continue....... (Read the below boxes also)

---

> ### Author Response · Authors · 2025-12-01
>
> Comparison of SSVR with SVQR: -
>
> In context of the SSVQR method comparison with the SVQR method, we claim the superiority only because the SSVQR model obtains the sparse solution and hence enable the feature selection in PI estimation. Our numerical results show that the SSVQR model performs similar to SVQR and superior in some cases in terms of the PCIP and MPIW values. However, SVQR model never obtains the sparse solution and hence cannot be used for feature selection.
>
>  Study of the non-crossing quantile:-
>
> The study of the non-crossing quantile becomes less relevant when you are estimation the PI with high confidence such as 95 % required for the real-world decision making.  The quantile functions of PI estimated for 95% coverage may cross each other only when the noise is not significant in data.  But, in such cases, the study of the PI becomes irrelevant and the decision can be made with the point and mean regression method.
>
> Experiments on standard benchmark datasets: -
>
> Our paper lists use the 12-13 benchmark datasets to highlight the efficacy of the SVM based UQ methods.
>
> Asymptotical properties of the SSVQR model: -
>
> I don’t know what is the source of the confusion. Please see the Lemma 3 of the (Takeuchi et al., Nonparametric Quantile Regression, 2005) that clearly shows that the pinball loss estimator minimized with any norm of regularization guarantees the asymptotically coverage for quantile estimation. The SSVQR model minimizes the L-1 norm with the pinball loss function.
>
> Reproducibility of numerical results: -
>
>  I can reproduce all the numerical results from given code but I should accept my mistake that  $\textbf{I should have provided the detailed readme file that can contains the instruction how to reproduce the results.}$
>
> I request you to consider my above arguments for possible clarifications of the confusions and give me the opportunity of the major revision.  Waiting for your reply.

---

> > ### Comment · Action_Editor_ucTd · 2025-12-02
> >
> > Dear Author,
> >
> > Thank you for your detailed message and for taking the time to further explain your perspective on the reviews and editorial decision for your submission to Transactions on Machine Learning Research (TMLR).
> >
> > I have carefully re-read your appeal alongside the referee reports and my original decision. I appreciate the clarifications you provide regarding your intended narrative, the role of SSVQR within SVM-based uncertainty quantification, and your comments on asymptotic properties, non-crossing quantiles, and reproducibility.
> >
> > However, after this further consideration, I must confirm that the decision to reject the manuscript remains unchanged.
> >
> > The main reasons are as follows:
> >
> > **Clarity and focus of the contribution**
> >
> > You emphasize that the primary contribution lies not in the SSVQR model itself, but in the broader use of SVM-based methods for various UQ tasks. The reviewers, however, consistently found the presentation difficult to follow, with the core methodological contributions and their novelty relative to existing SVQR/SVM work insufficiently clear. In your appeal, you largely reiterate the original framing rather than proposing a concrete restructuring or sharpening of the contribution that would address these concerns. This does not give sufficient confidence that a further revision would resolve the fundamental issues of clarity and focus.
> >
> > **Methodological justification and novelty over existing work**
> >
> > The decision letter noted that the genuine advances over existing SVQR-based approaches were not convincingly established. While you reiterate the importance of sparsity, feature selection, and stability, the appeal does not provide a more rigorous or better-delimited account of what is new, what is adapted, and how this goes beyond prior SVM/UQ methods. The ongoing ambiguity about what constitutes the “major contribution” reinforces the reviewers’ concerns rather than dispelling them.
> >
> > **Empirical evidence and strength of the experimental support**
> >
> > The reviewers questioned whether the empirical results truly substantiate the claimed advantages, especially given that the proposed method is not consistently superior to strong baselines. Your response restates that SSVQR is similar or better in some cases and highlights its sparsity, but does not address the broader concerns about experimental design, choice of baselines, and alignment of claims with evidence. A level of reworking would be needed here (e.g., redesigned experiments, stronger baselines, more careful claims) that goes beyond what can be resolved by another minor revision cycle.
> >
> > **Non-crossing quantiles and internal consistency of the evaluation**
> >
> > The reviewers’ concerns about non-crossing quantiles relate to the consistency between the assumptions underlying your evaluation (e.g., prediction interval width) and what is actually enforced by the models. Your appeal argues that this issue is “less relevant” for high-coverage intervals or in certain noise regimes, but it does not directly address the core methodological point: that metrics relying on non-crossing assumptions may be problematic when those assumptions are not enforced. This remains a substantive issue, not a minor one.
> >
> > **Asymptotic properties**
> >
> > Referring to an existing lemma in the literature is a useful pointer, but the reviewers requested a clearer and more self-contained treatment of the asymptotic properties in the context of your specific model. The appeal does not lay out how the conditions of the cited result are verified for SSVQR or how this would be made explicit and accessible in the manuscript. A future version would need to develop this more fully.
> >
> > **Reproducibility**
> >
> > The reviewers raised serious concerns about the completeness and consistency of the code and experimental setup. While you indicate that you can reproduce the results yourself and acknowledge the need for a more detailed README, this does not fully address the level of transparency and reproducibility expected for TMLR, especially given the strength of the reviewers’ doubts.
> >
> > Taken together, your appeal does not introduce new material or a sufficiently concrete and detailed revision plan that would alter the overall assessment: bringing the manuscript to the standards required for publication in TMLR would require a substantial reworking of the narrative, methodological exposition, empirical evaluation, and reproducibility materials, beyond what is feasible through another round of revision on this submission.
> >
> > For these reasons, I must reaffirm that the rejection decision is final, and we will not be able to consider a further revision of this manuscript at TMLR. I hope, however, that the points summarized above will be helpful as you substantially revise the work for submission to another venue.
> >
> > Thank you again for considering TMLR for your work.
> >
> > Sincerely,
> >
> > Action Editor